# Marine species and assemblage change foreshadowed by their thermal bias over Early Jurassic warming

**Carl J. Reddin** [1,2,3] ✉, **Jan P. Landwehrs** [3,4], **Gregor H. Mathes** [2,5], **Clemens V. Ullmann** [6], **Georg Feulner** [4] & **Martin Aberhan** [1]

A mismatch of species' thermal preferences to their environment may indicate how they will respond to future climate change. Averaging this mismatch across species may forewarn that some assemblages will undergo greater reorganization, extirpation, and possibly extinction, than others. Here, we examine how regional warming determines species occupancy and assemblage composition of marine bivalves, brachiopods, and gastropods over one-million-year time steps during the Early Jurassic. Thermal bias, the difference between modelled regional temperatures and species' long-term thermal optima, predicts a gradient of species occupancy response to warming. Species that become extirpated or extinct tend to have cooler temperature preferences than immigrating species, while regionally persisting species fell midway. Larger regional changes in summer seawater temperatures (up to +10 °C) strengthen the relationship between species thermal bias and the response gradient, which is also stronger for brachiopods than for bivalves, while the relationship collapses during severe seawater deoxygenation. At +3 °C regional seawater warming, around 5 % of pre-existing benthic species in a regional assemblage are extirpated, and immigrating species comprise around one-fourth of the new assemblage. Our results validate thermal bias as an indicator of immigration, persistence, extirpation, and extinction of marine benthic species and assemblages under modern-like magnitudes of climate change.

A suitable temperature is one of the most commanding habitat requirements for species at broad spatial scales[1]. Human activity has set isotherms on the move globally[2,3], leading to widespread shifts of marine species away from the tropics[4–6], with substantial repercussions for human well-being and ecosystems[7]. Although the focus tends to be on the decades up to 2100, warming will likely persist into the coming centuries[8] when climate change is anticipated to supplant land use change as the dominant driver of species extinction[9]. However,

species range shifts may indicate vulnerability to extinction[10]. Range shifts are expected to begin with an extension of their leading edge, as a species arrives into new habitat[4,11]. Trailing edge populations, meanwhile, may suffer performance decline as marine heat waves cause physiological stress[12], which can eventually lead to species extinction, both local (henceforth termed extirpation) and global (henceforth termed extinction)[10,13]. The proximity to a species' thermal niche edge should therefore indicate how a given population might

[1]Museum für Naturkunde Berlin – Leibniz Institute for Evolution and Biodiversity Science, Berlin, Germany. [2]GeoZentrum Nordbayern, Universität Erlangen-Nürnberg, Erlangen, Germany. [3]Alfred Wegener Institute Helmholtz Centre for Polar and Marine Research, Bremerhaven, Germany. [4]Potsdam Institute for Climate Impact Research (PIK), Member of the Leibniz Association, Potsdam, Germany. [5]University of Bayreuth, Bayreuth, Germany. [6]University of Exeter, Exeter, UK. ✉e-mail: carl.j.reddin@fau.de

react to warming[14,15], particularly for marine ectotherms, whose distributions tend to be closely associated to their thermal tolerances[16]. Future observations, including species extinctions, will provide greater predictive confidence. However, once a species has gone extinct, it is irretrievable, and climate-induced extirpations are already widespread[5]. Rather than waiting for climate-induced extinction to manifest, the rich fossil record has great potential to explore links between climate-induced extirpations and extinctions[17], especially given the recurring Earth system responses to a rapid addition of atmospheric $CO_2$[18,19].

Over long time scales, ecological assemblage change may be more frequent than remaining constant, allowing the fossil record to elucidate links between species range shifts, turnover, and extinction risk[20]. Climate change is consistently associated with species latitudinal range shifts and regional turnover across multiple marine taxa and time scales[21-23]. Global warming also fosters seawater deoxygenation in both the modern and the past[18,24], which can either make populations more sensitive to warming[25] or supersede the impacts of warming completely as anoxia[26]. However, the degree to which thermal preferences can be associated with the regional vulnerability of fossil populations, species, and assemblages during climate change remains unclear.

Thermal optima and tolerance limits may be conserved over millions of years[27] and can be estimated for a species based on its geographical distribution. Thermal optima can be compared among species at a location as a species temperature index (STI), or averaged to estimate an assemblage-level net preference at a location (community temperature index, CTI)[28,29]. Note that we define an assemblage, without any requirement of cohesion, simply as the species present in each spatiotemporal unit, throughout. An STI or CTI falling behind environmental change signifies a thermal bias[11,29], the difference between one or multiple species' long-term median temperatures and local ambient seawater temperatures. Thermal bias can indicate that populations are further from their respective species thermal optimum and closer to tolerance limits, potentially making the assemblage more vulnerable to species turnover than others[11,29]. In marine shallow-water fauna, assemblage thermal bias may even be more indicative of species loss than regional warming rates[29]. Therefore, thermal bias, STI, and CTI are valuable measures for species or community vulnerability under climate change. Although the thermal bias of fish and plankton species has been correlated with changes in their local abundance and occupancy[11,23,30], the wider validity of these metrics is rarely tested, especially at the assemblage level and their link to global extinction risk.

We expect that, (A) under warming, species' occupancy responses are ordered with respect to, and dependent on their thermal bias, the difference between the regional median temperature at a time zone and the species' thermal median. This means that species that immigrate to a region tend to have positive thermal biases − on average they have preferences for warmer temperatures than the ambient conditions −, while extirpated species and those going extinct tend to have negative thermal biases − a preference for colder temperatures than the ambient conditions (see Methods for precise definitions for occupancy responses). Finally, persisting species tend to have relatively intermediate thermal biases, with regional temperature change remaining within their range of tolerance. Species originating or going extinct could be considered the climaxes of this gradient of responses (response levels: originating = 1, immigrating = 2, persisting = 3, extirpated = 4, extinct = 5), which indicates how thermally well-adapted a species was to the new environment. (B) the temperature difference along this species response gradient is stronger with greater regional climate change, and for brachiopods than bivalves, the former being more sensitive to a given warming e.g. ref. 31. Finally, if a region is warmer than the thermal optima of many of its individual species inhabitants, a net negative thermal bias will emerge for the regional assemblage. We expect (C) that

assemblage-level thermal bias determines how an assemblage responds to warming. For instance, regional warming and an assemblage with an already negative thermal bias will lead to extensive assemblage-level change, characterized by high degrees of extirpation, immigration, and species turnover. Conversely, a region with little or no net assemblage thermal bias, or that is occupied by species with warmer optima than ambient temperatures (assemblage with positive net thermal bias), will change little under further warming. We test the above expectations using linear mixed effects models, with random effects to account for the same species being observed in different regions, and the same region being observed in different time zones. To guard inferences against changes in sampling intensity between time bins, we focus only on consistently well-sampled regions and two-timer species (sampled at least twice in consecutive time intervals), whose record of presences and absences may be less influenced by sampling fluctuations (see Methods). Extinctions and, for completeness, originations were identified by dataset-wide last or first appearance dates (LADs or FADs) of two-timer species.

Our study system consists of the well-sampled epicontinental seas of the north-western Tethys, before, during, and after an Early Jurassic extinction event[32] that is often considered as a modern analogue[18]. To help standardise sampling effort and scaling aspects, including approximating the spatial resolution of available climate model outputs, we identified major paleogeographical clusters of marine species occurrences and focus on these as discrete regions (Fig. 1). These regions, sampled in consecutive time intervals, are similar in area to regions used to investigate thermal bias of modern organisms[33]. Here, most marine benthic fossils are bivalves, brachiopods, and gastropods, whose species-level taxonomy is well-agreed, thus we focus on these three groups. Nevertheless, coastal taxa tend to be congruent in their diversity patterns[1], and richness patterns of marine molluscs, even limited to their most common species, serve as good indicators for other marine ectotherm clades[34]. Thus, our results may be valid for the wider marine macrobenthos. The late Pliensbachian to early Toarcian interval covers a transition from the coolest global temperatures of the Early Jurassic, potentially with polar ice sheets[35,36], through rapid global warming pulses with potential modern relevance[18], to stabilisation as a greenhouse climate with the warmest Early Jurassic global temperatures[37]. Comparing occupancy patterns in fossils over large distances requires that occurrences are time-binned to intervals that can be stratigraphically correlated over distance, typically the multi-million-year long geological stages. Here, we focus at a higher temporal resolution, the ammonite zone (mean = 1.1 myr), at which the climatic changes can be generalised into the following phases (Fig. 2A): little change between the two late Pliensbachian zonal means (termed the cold stasis phase); warming into the earliest Toarcian zone (termed the warming phase 1); further warming during the Toarcian Ocean Anoxic Event (T-OAE; termed the warming phase 2); an initial continuation of peak warm conditions before cooling slightly (termed the transitional phase, having the highest mean temperature); a stable, warm climate (termed the warm stasis phase; see Methods and Supplementary Methods 1). We used literature estimates of $CO_2$ concentrations[36,38], or geochemical proxies of seawater temperatures, for forcing the climate model, CLIMBER-X[39]. Our thermal bias calculations focus on modelled spatial variation in summer mean temperatures, because seasonal maximum temperatures may drive species extirpation during warming[13]. Using our models, we estimate occupancy responses at a modern-relevant magnitude of +3 °C regional warming.

## Results

### Thermal bias associated with sequence of species responses to warming

We observe palaeoecological change from one time step (ammonite zone) to the next and expect the transitional phase to capture many

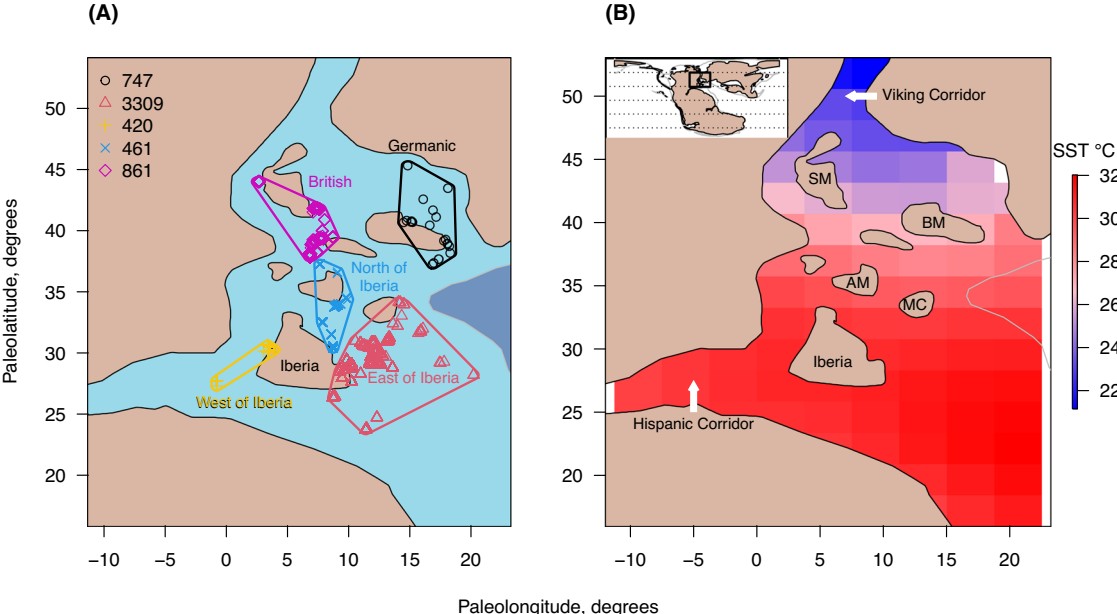

**Fig. 1 | Focal regions and example climate of the Early Jurassic, north-west Tethys. A** Symbols indicate all fossil occurrences between Margaritatus and Bifrons ammonite (time) zones, grouped into regions (coloured, and labelled) by hierarchical clustering based on occurrence paleocoordinates. Values in the legend are total number of occurrences per region. Paleocoordinates, maximum sea level coastlines (thin black lines), and deeper waters (dark blue, demarcated by −1400 m contour) were reconstructed according to Pliensbachian (185 Ma)

palaeogeography of the PaleoMAP model[80]. The landmass of Iberia is labelled. **B** An example of the utilized CLIMBER-X downscaled mean summer sea surface temperatures (SST) at the 185 Ma (Pliensbachian) paleoconfiguration and 750 ppm atmospheric $CO_2$. Global location shown as box in world map (inset top left) alongside lines of paleolatitude every 30 degrees including the equator. BM Bohemian Massif, MC Massif Central, AM Armorican Massif, SM Scottish Massif. Source data are provided as a Source Data file.

extirpations and extinctions from the rapid T-OAE warming within the Exaratum subzone. We therefore consider the species responses over the two warming phases and the transitional phase (Fig. 2A) to be warming-associated, having regional warming steps of $4.5 \pm 1.9\,°C$ (mean ± SD). Warming associated species' occupancy responses formed a gradient correlated with their thermal bias, as shown by climate phases separately in Fig. 2C–E and their mean in Table 1. This correlation was not supported during times of climate stasis (Fig. 2B, F). Negative (cool) thermal biases prevailed during warming associated phases (intercept in Table 1), especially visible in warming phase 2, when extinctions and extirpations were high (Fig. 2D). The gradient of species occupancy responses contrasted thermal biases of immigrating species from extirpated species. An immigrating species' mean thermal bias of $+0.5\,°C$ implies that their thermal optima approximately tracked ambient water temperatures, whereas extirpated species had more negative mean thermal biases of $-4.3\,°C$.

To assess how well observed thermal biases of species met linear expectations for the different species occupancy responses, we calculated confidence intervals using a linear mixed effects model, with thermal bias as the dependent variable. The thermal biases of most response levels approximated linear expectations: originating species had the warmest preferences (mean = $+1.8\,°C$); persisting species' thermal optima were significantly below local temperatures (mean = $-1.1\,°C$); species going extinct had most negative biases (mean = $-5.0\,°C$). However, the observed mean thermal bias for extirpated species of $-4.3\,°C$ fell below expectation (conditional mean = $-3.0\,°C$, 95% confidence intervals, CIs = $-4.1$ – $-1.9\,°C$). Over warming-associated phases, a species' thermal bias was a stronger predictor of its occupancy response than the magnitude of regional warming (Table 1). It explained 18% of the response variation whereas the magnitude of regional warming explained less than 1% in separate models ($R^2_{marginal}$ values).

To guard against criticism that originating and extinct species' thermal niches were pre-decided (e.g. because species going extinct in

time i can only have occurrences in the past relative to time i, when climates tended to be relatively colder in our study), we compare our results with regression results where extinction or origination responses were left out. The significance of this relationship was robust to whether origination and extinction responses were treated separately, pooled with immigrations and extirpations respectively, or excluded entirely (Fig. 2, Table 1, Supplementary Note 1, Supplementary Tables 1, 2). Our temperature estimates are dependent on climate model $pCO_2$ assumptions, and paleocoordinates are dependent on assumed paleogeographical reconstructions. Therefore, we also ran our analyses under different paleogeographical and $pCO_2$ assumptions. We observed the same general results (Supplementary Note 2, Supplementary Table 3), thus reaffirming our previous findings. We also assessed an alternative approach to control for sampling variation, which maintained the same general results (Supplementary Note 3, Supplementary Fig. 1). Finally, besides temperature, other widespread habitat changes might be expected to influence species regional occupancy, but we found no evidence for consistent impacts of habitat substrate change on species occupancy responses (Supplementary Table 4). While changes in water depth over the first warming phase coincided with an apparent immigration event into east of Iberia (Supplementary Note 4, Supplementary Tables 5, 6), the effect of thermal bias remained when this was accounted for, or, alternatively, if this region during the first warming phase was removed from analysis (Supplementary Note 4, Supplementary Table 4).

## Sources of variation in species responses to thermal bias

Support for the relationship between species' thermal bias and their occupancy response gradient varied by taxonomic group, and in time and space corresponding to regional warming and anoxia. Brachiopods were more affected by the magnitude of regional warming than bivalves (interaction term between clade and climate change magnitude in Table 1). Similarly, the effect of thermal bias on occupancy response was stronger in brachiopods than in bivalves, though

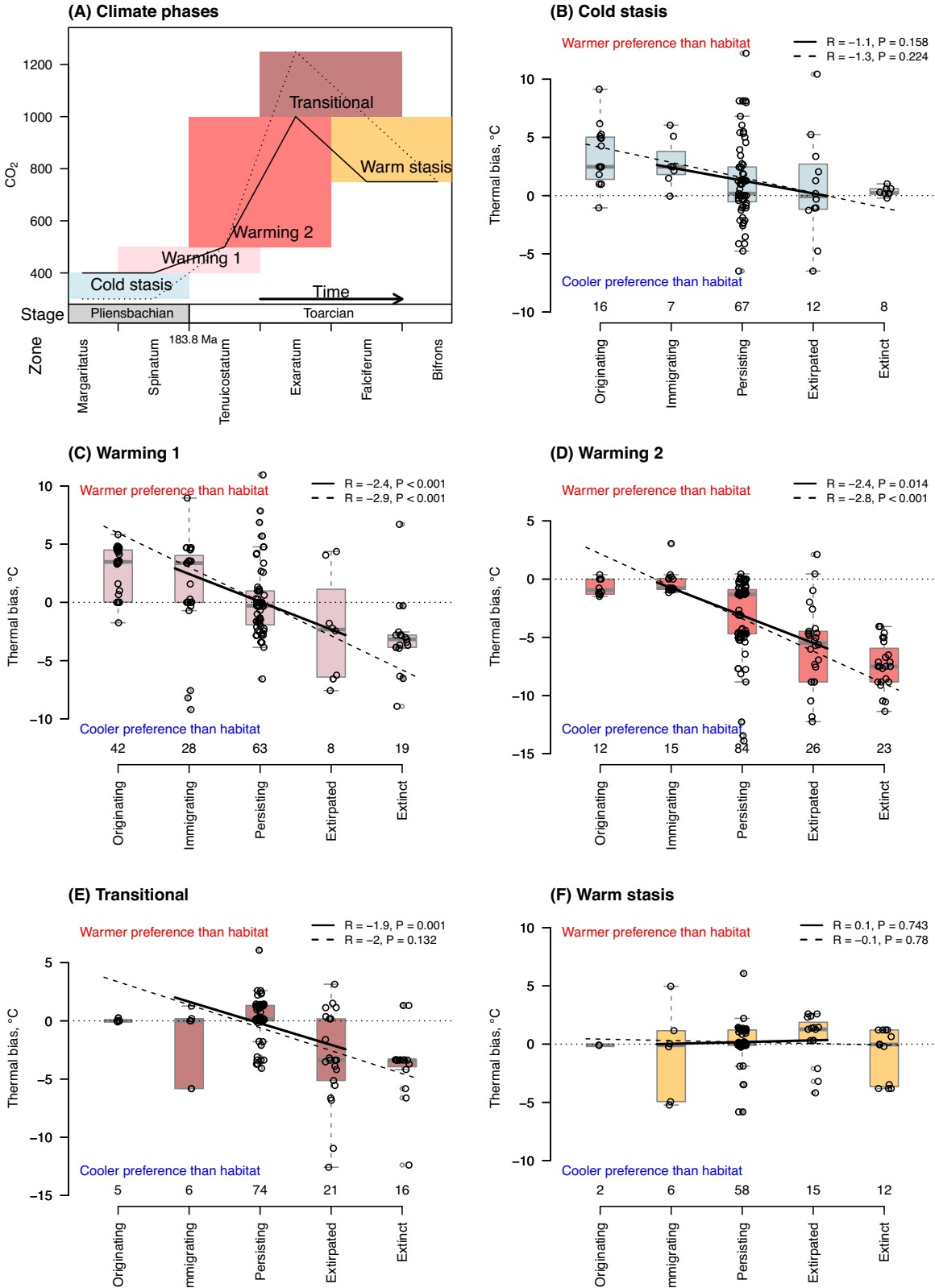

significance was marginal (interaction term between clade and thermal bias in Table 1). Greater regional warming strengthened the link between a species thermal bias and its occupancy response gradient, expressed by a steeper regression slope (Fig. 3). This was best supported when sample sizes were larger (i.e. more species), representing better sampling but also more oxic benthic paleoenvironments (Supplementary Fig. 2). In both phases of climatic stasis, there was no

significant relationship between thermal bias and occupancy response (Fig. 2). Support for the relationship was also weak to absent in the British basins and north of Iberia during the widespread bottom anoxia of the climate transitional phase, and throughout the Toarcian in the Germanic basins because of dwindling occurrences (Supplementary Fig. 3, Supplementary Note 4, Supplementary Table 6). This was despite the northern regions experiencing the largest warming

**Fig. 2 | Species' thermal bias is correlated with a gradient of occupancy responses over warming and transitional phases, but not over climate stasis phases. A** Summary of the climatic phases under study, with ammonite (sub)zone time bins on the x-axis. The solid line shows the main pCO$_2$ scenario, while the dotted line shows more extreme estimates. The stage boundary absolute timing has an estimated uncertainty of ±0.4 Ma[89]. The T-OAE occupies most of the Exaratum ammonite subzone. **B–F** Each panel shows two regressions: the solid line regressions run across immigrant, persisting, and extirpated species responses only; the

dashed line regressions run across all five ordered response levels. Regressions use random effects to nest regions within time zones. Circles show species responses with a small horizontal jitter to avoid overplotting of points against their thermal biases per region, the numbers of which are given along the x-axes, with box plots showing the medians and interquartile ranges, with whiskers extending to the furthest values within an additional 1.5x interquartile range. Significance testing was two-tailed, with exact *p*-values in **C** 0.0004 (upper) and 4.21E−14 (lower), and in **D** 8.35E−06. Source data are provided as a Source Data file.

## Table 1 | A species' occupancy response corresponds to its thermal bias during Early Jurassic climate warming and transition (the warmest) phases, and the effect of thermal bias is influenced by regional temperature change and clade membership

| Independent variable | Coefficient | S.E. | t-value | p-value |
|---|---|---|---|---|
| (Intercept) | 2.97 | 0.11 | 26.04 | 5.71e−88 |
| Thermal bias °C | −0.09 | 0.01 | −10.00 | 1.55e−08 |
| Regional temperature change °C | −0.03 | 0.03 | −1.19 | 0.268 |
| Clade [difference of Rhynchonelliformea vs. Bivalvia] | −0.05 | 0.07 | −0.74 | 0.458 |
| Thermal bias: Clade | −0.03 | 0.02 | −1.89 | 0.076 |
| Regional temperature change: Clade | 0.06 | 0.02 | 3.19 | 0.002 |

Coefficients from a mixed-effects multiple linear regression show the mean effect of one unit change in the independent variable (e.g. +1 °C) on occupancy response level, e.g. one level separates species that persisted from those that were extirpated. The intercept is the mean when all fixed effect variables equal zero, including the default setting of Bivalvia for the categorical variable clade. Interaction coefficients, indicated by colon symbols (:), show differences between bivalves and rhynchonelliform brachiopods in how occupancy responses are affected by thermal bias or regional warming. Here, extinction responses were treated conservatively as extirpations, and originations treated as immigrations, but the significance values are similar whether extinctions and originations were separate responses or whether extinctions and originations were removed entirely (Supplementary Tables 1, 2). Number of observations $n = 431$, comprising bivalves $n = 275$, brachiopods $n = 156$. 10 gastropods and 1 lingulid brachiopod observations removed here as too few to represent a clade. $R^2_{marginal} = 0.30$, the variance explained by these fixed effects alone; $R^2_{conditional} = 0.62$, variance explained by both fixed and random effects. Random effects nested species within region, and region within time zone. *S.E.* standard error

magnitudes (Supplementary Fig. 4; Germanic and British basins and north of Iberia), with climate scenarios estimating +7–11 °C over the two combined warming phases (up to +16 °C in a less likely pCO$_2$ scenario). Following our climate modelling, the British and Germanic (paleo)regions were always the coolest, with initial summer mean temperatures of 17–21 °C (Supplementary Fig. 4). Other regions warmed +6–9 °C over the two combined warming phases (up to +12.5 °C north of Iberia in a less likely pCO$_2$ scenario, Supplementary Fig. 3).

### Assemblage-level thermal bias and responses

Faunal responses to modern climate change are often averaged and projected to the future at the level of assemblage e.g. refs. 28,29. To mirror this approach, we take the mean thermal bias over species regionally present before a climate change and correlate it with the proportion of a given assemblage that were subsequently extirpated or went extinct, the proportion of species added via immigration or origination, and the overall turnover. Over all climate phases combined, a mixed effects model showed that assemblages increased thermal bias as the ambient temperature changed, adding −0.41 °C thermal bias (95% CIs = −0.77−−0.05 °C) for each degree of warming, rather than maintaining perfect thermal equilibrium (i.e. 0 °C thermal bias added per degree warming) or not responding at all (each degree warming adds −1 °C to thermal bias). Relationships were not different if assemblage thermal bias was weighted towards cool- or warm-adapted

members of the assemblage (Supplementary Note 5, Supplementary Table 7).

During the climate warming and transition phases, a negative (cooler) assemblage thermal bias consistently increased the proportions of species going extinct, being extirpated, or subsequently immigrating or originating. However, the small assemblage-level sample size meant that only the correlation with originations was significant (+1.3% in the subsequent assemblage per −1 °C assemblage thermal bias, 95% CIs = 0.6−2.0%; Fig. 4). The small sample size consistently increased the proportions of species changing, either going extinct, being extirpated, or subsequently immigrating or originating. The magnitude of regional warming was significantly correlated with an increase in immigrating species as a proportion of the subsequent assemblage (+8.5% per 1 °C increase in water temperature, 95% CIs = 4.2−12.8%; Fig. 4). The correlation between the proportion of species being extirpated and the proportion of species going extinct increased from R = 0.40 (*P* = 0.058) across all climatic phases to R = 0.73 (*P* = 0.006) during the climate warming and transition phases (mixed effects models of standardised variables). This reflects the influence of regional warming magnitude on the occupancy response gradient of species (e.g. Figure 3) at the assemblage level. Meanwhile, the proportions of immigrating and originating species in a new assemblage were moderately correlated, both during the climate warming and transition phases (R = 0.49, *P* = 0.092) and across all phases (R = 0.45, *P* = 0.033; mixed effects models of standardised variables). At the assemblage level, no significant effect of changes in habitat substrate or water depth was found (Fig. 4, Supplementary Note 4).

Regional temperature changes ranged from 2 °C cooling to +10 °C warming (i.e. x-axis of Fig. 3). The relationships in Fig. 4 estimate that, at a modern-relevant regional seawater warming of +3 °C, 4.74% (95% CIs = 0.03−9.45%) of an assemblage's pre-existing benthic species were extirpated and 25.5% (95% CIs = 12.5−38.4%) of a subsequent assemblage newly immigrated (Supplementary Note 6). As is typical for paleobiology, the context of the data underlying our estimates may differ substantially from modern settings, but, with critical evaluation, paleobiological insights provide unrivalled opportunity for validation of theory (see Discussion).

## Discussion

### Does a species thermal bias predict extirpation under climate change?

Regional species extirpation is often correlated with climatic change[6,13,28] but considering climate change relative to a species' thermal niche leverages additional information to assess population and assemblage vulnerability[29,33]. We present empirical evidence from the fossil record that immigration, persistence, extirpation, and likely the extinction of species form a response gradient during warming linked to how closely a species' thermal preferences fit regional conditions, as estimated by their thermal bias. This provides some validation that, as often hypothesised in ecology[11,15,29], the position of a population within its species' thermal niche is a useful attribute to predict its likely occupancy response to warming. However, thermal bias alone is likely insufficient to predict a species response to warming (e.g. $R^2 = 18\%$), with magnitude of regional warming and the species'

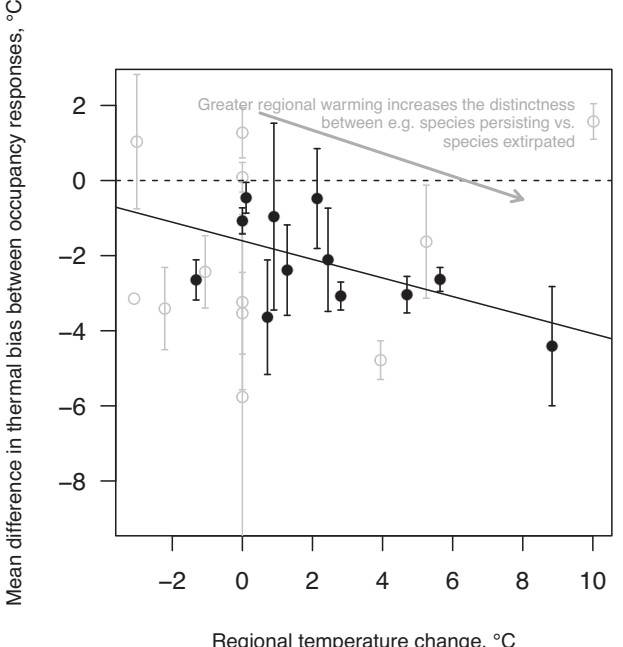

**Fig. 3 | Higher magnitudes of regional seawater warming drive a stronger link between thermal bias and species occupancy change.** Each point ($n = 25$) shows the mean difference in thermal bias between species occupancy response levels (°C value on the y-axis; i.e. each is a linear regression coefficient with its standard error plotted as error bars; see Source Data file for summary statistics for each) for a single region and time against the temperature change magnitude for that region and time (x-axis). Standard errors were used for inverse variance weighting of least squares regression. The weighted regression for the regions and times with at least 20 species (filled circles $n = 13$) is shown by the solid black line, R = −0.25 (95 % confidence intervals or CIs = −0.49–−0.01, $P = 0.044$), while the slopes of the other lines were insignificant (see Supplementary Fig. 2). Extinct species' occurrences are here merged with those of extirpated species and originating species occurrences are merged with those of immigrating species (though the same result was achieved treating these groups separately, R = −0.18, $P = 0.0498$, with threshold = 20). Significance testing was two-tailed. Source data are provided as a Source Data file.

taxonomic grouping as bivalve or brachiopod explaining additional variation (here, altogether $R^2 = 30\%$). The focal spatiotemporal extent and species here were well-sampled, supporting the validity of observed responses such as extirpation. However, highly variable thermal biases of individual species across responses likely indicates the influence of other niche aspects[33] on responses of mid-latitude bivalve, brachiopod, and gastropod species, as well as persistent sampling biases. In some times and regions, seawater anoxia was likely responsible for species absence from assemblages, rather than temperature. The otherwise consistent temperature–response relationship supports cautious use of habitat suitability models (also known as species distribution models, the spatiotemporal projection of ecological niche models) to predict extinction risk based on temperature change as a statistical tendency[40]. Ideally, especially at finer spatial scales, additional variables such as biotic interactions will support a more comprehensive approximation of marine species' niches and responses to change[33].

For simplicity and parsimony of assumptions, especially given the large time scales, we fit a linear relationship between a species' thermal bias and its regional occupancy response, from immigration, through persistence, to extirpation (detailed in ref. [10]), and potentially extinction. This is intuitive for species able to disperse over an unrestricted thermal landscape, which was relatively well supported. However, sampling was bookended between approximately 15 and

34 °C (Supplementary Fig. 5), with geographical barriers to the north (discussed below). Ideally, thermal preferences (e.g. STI values) should be estimated over a species' entire geographic distribution[30], though the patchy fossil record means this is rarely possible. Heterogeneity in habitat variables other than temperature are likely to dominate at finer scales than those studied here, with our regions being 1000 s km across. As climate changes, species distributions should move through a region, following shifting isotherms according to their thermal tolerance, other habitat variables permitting[10]. Species responses are likely to have additional dimensions that also correlate with thermal bias, such as stunted growth in persisting populations[41,42], and larger body sized species being regionally replaced by smaller, opportunist species[43,44].

The influence of thermal bias on occupancy response was more pronounced in rhynchonelliform brachiopods than bivalves, the latter likely being less vulnerable to extinction during geologically rapid warming[31,43,44]. Thermal performance has ecophysiological underpinnings[25,45,46], with some organisms having more limited physiological adaptations. Evidence is mounting that different ecophysiological adaptations among taxa lead to different performance outcomes, including extinction risk[25]. However, quantitative comparisons of the thermal performance of brachiopods and bivalves are scarce[47]. Our results therefore support the view that ecophysiology predisposes some taxa to greater species immigration and extirpation at multiple scales, with their extinction risk being predictable via habitat loss[46,48,49], including oxic habitat. Groups vulnerable to warming, such as bony fish[4,25], may thus be more likely to show strong range shift responses as they rely on stressor escape, where habitat permits, rather than tolerance. Other vulnerable groups, such as reef corals, may be more restricted in their rate of habitat tracking (though see ref. [22]). Combining physiological principles and environmental factors[46] may aid understanding of the pressures regional warming will place on species via identification of vulnerable clades or traits, alongside spatial projections of their habitat loss[48–51].

## Linking climate-driven range shifts to extinction risk

Colonising newly suitable habitat may allow species to avoid climate-induced extinction[13,16]. Therefore, marine fauna are expected to consistently trace their thermal preferences during climate change[21]. An occupancy response gradient in line with a species' thermal bias may be a null expectation for a warming habitat[10,15], leading species with particularly negative thermal biases to be vulnerable to local and global extinction. However, there are modern examples of how disequilibria between ambient temperatures and a population or assemblage can be stable rather than owing to species dispersal failure, especially when observed at finer spatiotemporal scales than sampled here[15,29]. Even at our scales, we observed a large variation in thermal bias for persisting species, suggesting either that finer scale temperature differences played a substantial role in permitting species to persist (i.e. thermal refugia), or that many species were eurythermal. Nevertheless, our finding that regional warming increased the slope of the relationships between thermal bias and response implies that greater magnitudes of warming on average increase the cost of disequilibria between species and climate. Furthermore, the overwhelming negativity of thermal biases across responses during warming phase 2, which coincided with the highest ratio of extirpations and extinctions to persisting species, may also have been amplified by the context of short-term warming on-top-of legacy warming, which can increase extinction risk[52].

The largely overlapping thermal bias values for species being extirpated and going extinct, as observed here during warming and transitional phases, may betray a cause of extinction. The mean thermal bias for extirpated species fell below the expectations of our linear model and instead fell within 95% confidence intervals for species going extinct, while observed thermal biases for immigrating and

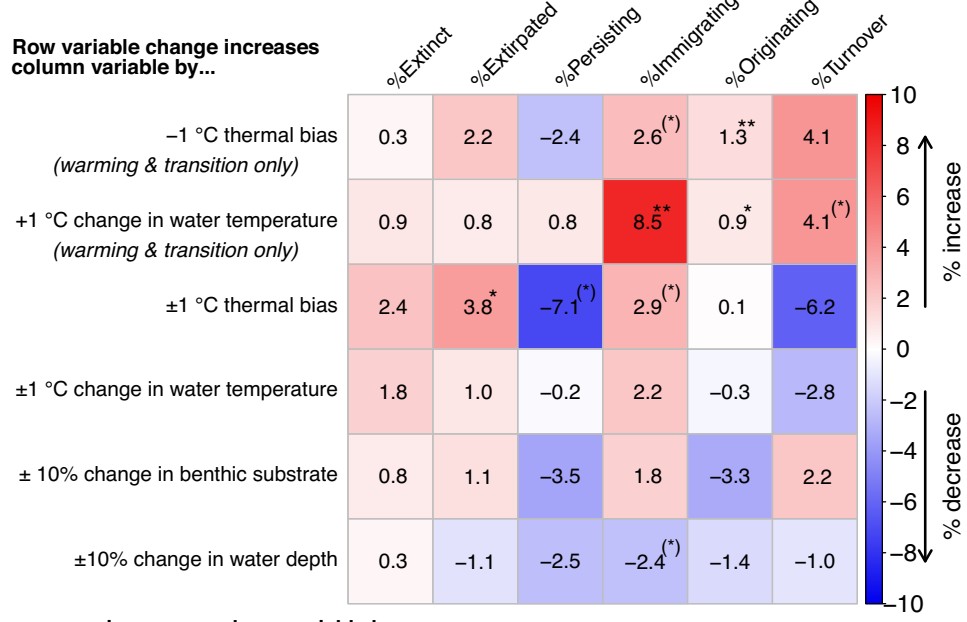

**Fig. 4 | How thermal bias at the assemblage level related to future assemblage change in %, versus whether direct descriptors of environmental change magnitude were more useful.** Figure should be read from row to column, with the intersecting cell showing the effect in %-change in the column variable and its significance. For example, reading the first row: for each degree Celsius of thermal bias below ambient water temperature, the proportion of originating species in the new assemblage increases by -1.3 % ($P = 0.001$). The empirical range of regional climate change was −2 to +10 °C. The cells show unstandardized coefficients from nested random effects linear models by colour (see colour scale). ** is $P < 0.01$, * is $P < 0.05$, (*) is $P < 0.1$. The first two rows have a unidirectional hypothesis between temperature change and response; $n = 12$ assemblages, 4 regions nested in 3 time zones (Germanic basins responses were unavailable). Other rows cover all five time zones with a bi-directional hypothesis between change and response; $n = 22$ assemblages, 5 regions nested in 5 time zones (Germanic basins responses were unavailable for three time zones). The last two rows are %-change of the occurrences per zone and region that are categorized as primarily carbonate lithology or deep depositional environment, indicating larger changes in the sampling of these habitat types within a region. Significance testing was two-tailed. Source data are provided as a Source Data file.

originating species aligned well with linear expectations. We suspect that the thermal bias values for extirpated species were valid but we observed more extinctions than expected, given their modest thermal bias values. Given the anoxia of northern waters of the north-western Tethys, especially during the T-OAE (discussed below)[26,53], we suspect anoxia, rather than temperature, dictated habitat suitability. This may have left many species with thermal bias values suggestive of extirpation with no oxic habitat left to disperse to, leading to their extinction responses. Poor dispersal capabilities and/or dispersal barriers can lead to a species' failure to reduce its population thermal biases by shifting distributions, thereby shrinking its geographical range[54], and making it vulnerable to global extinction[49,55]. For greater understanding of context dependence, mechanisms of habitat tracking and their limitations should be explored across multiple intervals with changes in climate, sea level, and geography refs. 51,55.

### Influence of anoxic waters and terrestrial runoff

The well-sampled, oceanic-influenced regions north and east of Iberia best supported the correlation of species' thermal bias values with a gradient of their responses, where any early Toarcian deoxygenation prior to the T-OAE[56] apparently did not preclude a signal of thermal bias. Analyses of well-oxygenated environments such as outcrops from the south-west of Europe[53] implicate early Toarcian warming as the main regional driver of species loss, changes in bivalve-brachiopod assemblage structure, and their body size[43,44,57]. During peak T-OAE (warming phase 2 into the transitional phase), support for the link between thermal bias and the occupancy response gradient dwindled in the Germanic and British regions alongside the number of fossil occurrences. Although aquatic deoxygenation can amplify the influence of warming on ectotherm performance[25,58], bottom water anoxia

is likely to supersede the ecological influence of warming. Several regions during the T-OAE are characterized by black shale deposition, where hypoxic and anoxic waters have long been associated with faunal turnover and extinctions[26]. Accordingly, benthic macrofaunal recovery only began after seafloor ventilation resumed, and remained incomplete in the British region by the end of our study[26,59]. During the T-OAE, the northern waters may have essentially been unavailable as habitat for species tracking their thermal and oxic niches, forming a dispersal barrier. This may exemplify how species ranges can be compressed as they trace thermal preferences[48,49]. Although fully marine (see Supplementary Table 8), the more restricted northern waters likely had greater terrestrial influence, such that bottom-water anoxia was probably dependent on stratification[53] and productivity, as nutrients were delivered from warming-enhanced weathering[60], rather than simply temperature-dependent deoxygenation. The HadCM3 model estimated slightly lower salinity in the Germanic and British basins also[53,61], ranging between 33.3 and 34.6 ppt across scenarios, than the other regions, while salinity was always highest east of Iberia, ranging between 34 and 35.6 ppt (Supplementary Note 7). The semi-enclosed setting, especially of the Germanic and British basins, also likely increased the influence of local processes that global models are unlikely to capture, with the reality likely being warmer and more seasonal than estimated by our models[62]. Regional climate models suggest that, although wind stress was likely southward[61], a clockwise gyre over the European epicontinental shelf had mostly weakened by the time it reached the northern shelf, making northern regions sensitive to local stratification[53]. Alongside changes in sea-level-dependent seafloor ventilation[59], water density differences from freshwater input likely encouraged stratification[53,61,63]. While modern oxygen minimum zones continue to spread[24], our results show how regional-scale

physical and biochemical processes can complicate the predictability of species and assemblage responses to temperature change[51].

Besides temperature and salinity, other broad-scale habitat requirements for a benthic species include suitable water depth and substrate conditions, which also dictate the conditions under which a species can be sampled. The northern regions were the only ones dominated by siliciclastic substrates, which could have blocked the immigration (alongside anoxia, see previous paragraph) of carbonate-affinity species. The largest and most consistent non-temperature change occurred at the Spinatum-Tenuicostatum ammonite zone transition, when substantial sea-level rise[36] led to increases in the frequency of deep habitat occurrences from 20–50% to 90–96% per region. However, a species' thermal bias remained highly significantly associated with its occupancy response through different statistical treatments that explored the importance of this spatiotemporal scenario (see Supplementary Tables 4, 5). Being 100 s km across, our regions tended to cover substantial substrate and depth variation, such that finer scale analyses may be needed to detect the influence of habitat variables other than temperature (see next section), including biotic interactions. Our focus on two-timer species also emphasised longer-term changes of the more common and better-preserved species, of which our analyses support temperature change being a key driver at broad spatial scales.

## Temporal and spatial scaling

Temporal and spatial resolution or units in our study were ~1 million years and ~2000 km respectively, which need appreciation to compare our results with other studies and drivers. Finer scale variations, varying within the units above, were averaged out, such as the warming at the Pliensbachian–Toarcian stage-boundary[64], despite permanent palaeoecological changes such as extinctions remaining from short-term pulses. In contrast to the myriad of factors influencing a species occurrence at fine spatial scales, where biotic interactions may be especially important[33], at broader scales climate is expected to be one of the dominating factors[15,65]. Significant effects of thermal bias have been assessed for modern assemblages at spatial scales from surveyed sites[11] to biogeographic ecoregions, more similarly sized to our regions[29,65]. At intermediate spatial scales, Flanagan et al.[33] found larger thermal biases of fish assemblages over decadal scales than inter-annual scales, which might encourage expectations that marine communities rapidly maintain equilibrium with temperature[33]. At much longer time scales and with spatially coarse temperature estimates, our data also supported a general equilibrium between the mean of species thermal optima in an assemblage (CTI) and environment temperatures (Supplementary Fig. 6). However, geographical context affected observations of thermal equilibrium in a study of planktonic foraminifera over thousands of years: mid latitude assemblages tracked climate change by turnover, but decreasing assemblage turnover at high latitudes under warming and low latitudes under cooling led to observations of assemblage thermal bias[23]. Regions of high climate velocity, such as the tropics and poles, are likely to demand faster species' niche-tracking than lower climate velocity regions, which is more likely to push populations of multiple species nearer to their thermal tolerance limits[49]. However, increasing thermal bias may only increase extirpations and extinctions when changes exceed species' recent climatic experience[52]. Temporal resolution is not a problem per se for the application of palaeontological insights to modern issues[17], but it limits the mechanisms, which may or may not be applicable to modern scenarios, for which we can observe evidence. Future work should be directed to understanding the mechanisms underlying observed palaeontological patterns and the transferability of those mechanisms to modern climate change and the current biodiversity crisis[5,17].

Regional warming estimates for the northwestern Tethys during individual warming phases were $4.5 \pm 1.9\,^\circ C$ (mean ± SD; x-axis of

Fig. 3). These cover magnitudes similar to end of the century forecasts under high emissions scenarios (RCP 8.5) for some modern regions, such as $+3\,^\circ C$ for the North Sea[66], although at very different rates of change. At $+3\,^\circ C$, our model already expects regional benthic species extirpations and especially immigrations to be considerable (4.7%, CIs = 0.03–9.5% and 25.5%, CIs = 12.5–38.4%, respectively; Supplementary Note 6). These extirpation and immigration values are similar to projections of a paleo-validated biodiversity model for the shelf seas of Europe by 2100[67]. Although relating our results to modern warming ignores the very different time scales (=observed rates of change), the loss of a species' thermally suitable habitat can respond directly to the magnitude of warming, regardless of its rate of warming. This was shown by simulated patterns of high latitude extinctions during hyperthermal events, while low latitude extinctions were more rate-dependent[49]. Rates of ancient climate changes, although variable, may have been sufficiently slow for most species to track habitat availability. However, the extremely rapid anthropogenic rates of change are likely to divide response severity between species with greater and lesser dispersal abilities[49]. This may be especially the case in the tropics where climate velocities are highest[68], leaving paleobiological extrapolations most likely as underestimates.

Rare species, both range-restricted or wide-ranged but locally uncommon, are unlikely to be represented in the fossil record and thereby in our analysis. If rare species are at higher extinction and extirpation risk or tend to have narrower thermal tolerances, including them can be expected to raise the overall magnitude of assemblage change above our predictions. Again, this implies that inferences based on paleobiology will tend to give underestimates of whole community responses.

In summary, we show a distinct relationship between the thermal suitability of Jurassic bivalve, gastropod and brachiopod species for their occupied region and their occupancy changes in that region during warming. Thermal bias, i.e. the mismatch between a species thermal optimum and ambient water temperature, provides more information than the magnitude of regional warming alone and thus can be a stronger predictor of species extirpation, persistence, or immigration. Furthermore, species-level responses aggregated to substantial assemblage-level responses. Temperature-focused models may be less effective at finer (more local) spatial scales, where additional habitat variables may become more important, and in semi-enclosed coastal waters, which may be more inclined to anoxia. Predictions may be further refined by species-specific modelling and using climate models that handle processes at regional or finer scales, such as tidal mixing, where permitted by reliable, high resolution paleogeographic reconstruction. Our results support that greater magnitudes of warming tend to increase the cost of disequilibria between species and climate, increasing the rate of extirpation and, if thermal habitat is not replaced elsewhere, extinction. At the assemblage level, ambient warming was most clearly linked to increased species immigration. Given potentially unprecedented modern rates of global warming[69,70], paleobiology likely presents conservative warnings of future changes in marine species' regional occupancy.

## Methods
### Study interval and region
We focus on the climate changes from the cool late Pliensbachian to the warm early and middle Toarcian (Early Jurassic), covering the hyperthermal Toarcian Ocean Anoxic Event (T-OAE), when some ocean basins became anoxic[26]. We used the finest temporal resolution that is regionally consistent for our occurrence data, the ammonite zone (the Serpentinum Zone was further split into Exaratum and Falciferum subzones; Table 2; mean 1.1 myr). We estimated local climates at this resolution using published temperature proxies (particularly Müller et al.[71] and Ullmann et al.[72]; more detail in Supplementary Methods 1). After the cool, low-$CO_2$ late Pliensbachian Margaritatus

**Table 2 | pCO₂ scenarios used for modelling climates over time steps of ammonite (sub)zones, from the late Pliensbachian (Margaritatus Zone) into the middle Toarcian (Bifrons Zone)**

| Ammonite (sub)zone | Main pCO₂ scenario (ppm) | Secondary pCO₂ scenario (ppm) | Notes |
|---|---|---|---|
| Bifrons | 750 | 750 | Outputs should be warmer than Tenuicostatum[a] |
| Falciferum | 750 | 1000 | Outputs should be warmer than Tenuicostatum[a] |
| Exaratum | 1000 | 1500 | 1000 as low estimate. 1500 as peak estimate |
| Tenuicostatum | 500 | 500 | Secondary scenario with Pliensbachian map |
| Spinatum | 400 | 300 | 300 as cold estimate |
| Margaritatus | 400 | 300 | 300 as cold estimate |

See Supplementary Methods 1 for more detail on determining the pCO₂ scenarios.
[a]Following oxygen isotopes covering our temporal range in Müller et al.[71] or Ullmann et al.[72].

and Spinatum zones, the early Toarcian was associated with the release of greenhouse gases from the intense volcanism of the Karoo-Ferrar magmatic province[37,73,74]. Emplacement of the Karoo-Ferrar large igneous province occurred over ~9 million years between 183.4 and 176.8 Ma, with bulk magmatism occurring from -183.4 to -183.0 Ma, coinciding with the T-OAE[75]. Note that we consider the T-OAE to be equivalent in time to the well-known negative excursion of carbon isotopes (see Erba et al.[76] for discussion and alternative definitions). Analyses of thallium isotopes suggest that global marine deoxygenation of ocean water started sooner[56], alongside rapid, short-lived warming across the Pliensbachian/Toarcian boundary[73] at -184 Ma[77]. The Tenuicostatum Zone of the earliest Toarcian remained on average warmer than the late Pliensbachian. Further warming in the T-OAE proper of the Exaratum subzone, possibly as the consequence of a rapid release of thermogenic and/or biogenic methane adding to the volcanic $CO_2$ release, is associated with the main extinction phase[38,78]. After this peak of warming and $CO_2$ concentrations, the Falciferum subzone represents a transitional climate, starting warm but later cooling to a level warmer than the Tenuicostatum Zone[71], which is maintained into the Bifrons Zone.

Our regional focus, which offers the most densely sampled area during this time, follows a roughly north-south trending oceanic transect from Scotland via the western European epicontinental sea to north-western Tethys including Morocco, Tunisia, and Algeria (Fig. 1). Terrestrial influence (nutrients, turbidity, freshwater input) was higher in northern, more restricted water bodies, especially the Cleveland Basin[26], with less mixing and less oxygenation of bottom waters[61,79]. This is particularly expressed during the Exaratum subzone (T-OAE proper) when sites in England and Germany are dominated (though not completely) by hypoxic to anoxic sediments, while other basins were less affected by deoxygenation e.g. refs. 44,53.

**Seawater temperature maps**

pCO₂ scenarios per ammonite zone were either allocated directly, where pCO₂ estimates were available (Tenuicostatum and Exaratum (sub)zones)[36,38], or indirectly based on approximating relative temperature change estimates. In particular, Müller et al.[71] and Ullmann et al.[72] traced relative temperature change via oxygen isotopes of well-preserved brachiopod shells over our whole temporal duration. Temperature changes output by the CLIMBER-X climate model were then checked against proxy temperature changes at the appropriate paleocoordinates and water depths (Supplementary Note 8). Secondary pCO₂ scenarios were based on maximum possible temperature changes (Table 2; see Supplementary Methods 1 for a wider discussion of the evidence).

We ran equilibrium climate simulations at fixed pCO₂ scenarios using the CLIMBER-X Earth-system model[39]. CLIMBER-X is particularly useful as a fast and flexible paleoclimate model and provides simulated temperatures in the ocean and atmosphere on a 5° × 5° horizontal grid, among other parameters. Early Jurassic boundary conditions were represented by a reduced solar constant (1340.5 W/m²). For the model

palaeogeography, we used the bathymetric topography of Kocsis and Scotese[80], which matched the coastline to marine occurrences in the Paleobiology Database (see below), primarily using the Toarcian map (180 Ma) and secondarily using the Pliensbachian (185 Ma). Deep sea-floor depth was set to −3700 m, marine continental shelf to −200 m, and land to +200 m. Local shelf features are not well represented in these reconstructions and the coarse resolution model results are not expected to be perfect, but we expect the derived temperature estimates to give more accurate estimates of relative thermal preference than paleolatitude. We also downloaded the sea surface temperature maps simulated for 180 and 185 Ma with the HadCM3 model, though these were limited to pCO₂ scenarios of 560 and 950 ppm[81]. Despite being affected by similar limitations, HadCM3 is a more complex and highly resolved model than CLIMBER-X, and its outputs were used as a benchmark. The similarity of seawater temperature features produced by the two models supported the downscaling of the July mean temperature maps from CLIMBER-X to the finer spatial resolution of the HadCM3 maps via bilinear interpolation. Alternative downscaling via nearest neighbour resulted in no practical improvement in spatial resolution.

In general, correlations between the two models were high (Rho ≥0.8) with a root mean square error (RMSE) that increased, as expected, as the modelled pCO₂ scenarios deviated (other points for consideration are in Supplementary Methods 2). When comparing different model outputs (Supplementary Fig. 8), we used multiple ways to examine the differences between two sea surface temperature (SST) maps. Subtracting the two SST maps showed the differences as a map, which can indicate areas where one or both are erroneous (Supplementary Fig. 7). The correlation (Rho) between two maps can highlight differences in trends, even if the absolute differences are not large, if the correlation is very different from 1 (Supplementary Tables 9, 10). Root Mean Squared Error (RMSE) is also often provided to quantify the deviation of two spatial fields (Supplementary Tables 9, 10). RMSE is the standard deviation of the differences between the two maps (the residuals), measuring how spread out these residuals are. In other words, it shows how concentrated the data are around the line of best fit. There is a direct relationship with the correlation coefficient. For example, if the correlation coefficient is Rho = 1, the RMSE will be 0, because all the points lie on the regression line (and therefore there are no errors). However, a correlation coefficient and RMSE can pick up on different aspects of variation. For instance, there can be a good correlation even though there is a large offset between the two data sets giving a large RMSE. RMSE can be normalized to the unit of the dependent variable to facilitate comparison between values. While both models have an equilibrium climate sensitivity well within the range of CMIP6 models[2], the HadCM3 model is more sensitive than CLIMBER-X and generally yields higher temperatures at elevated $CO_2$ levels.

**Species occurrence data**

On 24th May 2022, we downloaded marine-only occurrences of bivalves, gastropods and brachiopods from the Paleobiology Database

(PaleoDB, https://paleobiodb.org/), representing benthic assemblages, and binned them to stratigraphic stages using R package divDyn[82]. Occurrences initially had to be accepted at least at the genus level (to allow checking whether their identified name could be vetted here as an accepted species), but our analyses used species-level occurrences. They also required modern geographical coordinates, which were used for paleogeographical rotation into paleocoordinates (see below) and for removing occurrences outside the north-west Tethys by a bounding box around modern Europe, east-west from Turkey to Portugal, and north-south from Scotland to the Mediterranean coast of Africa. Confidently identified species names that were taxonomically unaccepted by the PaleoDB underwent automatic checks for spelling mistakes. Of these, persistent unaccepted species names of the Pliensbachian and Toarcian were then taxonomically vetted by M. Aberhan to catch more accepted species occurrences and prevent artifacts in geographic distribution patterns, such as synonymous species names (Supplementary Data 1). To achieve ammonite (sub)zone temporal resolution, we explored the PaleoDB download columns named early_interval, zone, and stratcomments for temporal resolution information, especially ammonite zone or subzone allocation (Supplementary Data 2). The references of some data-rich entries were investigated manually for lacking temporal, paleoenvironmental or lithological information (see R code in Zenodo repository). A separate, global dataset was used to establish species' First and Last Appearance Dates (FADs and LADs), ideally at ammonite (sub)zone resolution, within the Pliensbachian and Toarcian stages. All references that contributed data for this study are listed in Supplementary Data 3.

Determination of species' thermal preferences may be confounded if species have significant affinities for particular substrate or bathymetric paleoenvironments. The diverse substrate or bathymetric categories listed in the following paragraph were combined using dataset 'keys', using lists 'lith' and 'bath', in R package divDyn. Only "unknown" values were uncategorized. Environmental affinities to carbonate or siliciclastic substrates, or deep or shallow depths, were tested for by using binomial tests with alpha = 0.1 (function affinity() in divDyn)[82].

The following depositional lithologies were categorised as 'carbonate' habitats: "carbonate", "limestone", "reef rocks", bafflestone, bindstone, dolomite, framestone, grainstone, lime mudstone, packstone, rudstone, floatstone, wackestone. The following depositional lithologies were categorised as 'siliciclastic' habitats: "shale", "siliciclastic", "volcaniclastic", claystone, conglomerate, mudstone, phyllite, quartzite, sandstone, siltstone, slate, schist. The following depositional environments were categorised as 'shallow' water habitats: coastal indet., delta front, delta plain, deltaic indet., estuary/bay, foreshore, interdistributary bay, lagoonal, lagoonal/restricted shallow subtidal, marginal marine indet., open shallow subtidal, fluvial-deltaic indet., paralic indet., peritidal, prodelta, sand shoal, shallow subtidal indet., shoreface, transition zone/lower shoreface, intrashelf/intraplatform reef, reef, buildup or bioherm, perireef or subreef, platform/shelf-margin reef. The following depositional environments were categorised as 'deep' water habitats: basinal (carbonate), basinal (siliceous), basinal (siliciclastic), deep-water indet., deep subtidal indet., deep subtidal ramp, deep subtidal shelf, offshore, offshore indet., offshore shelf, slope, submarine fan, offshore ramp, basin reef, slope/ramp reef.

Temperature estimates were sampled per taxon occurrence from modelled seawater temperature paleogeographical maps from 180 Ma (Toarcian, used primarily) and 185 Ma (Pliensbachian, used secondarily) separately. This avoided switching between maps in the same analytical time series, which could result in a sudden, artificial shift in paleocoordinates and influencing the thermal bias. Accordingly, we reconstructed coordinates and coastlines using the rgplates interface[83] to Gplates v2.3[84] to both Toarcian and Pliensbachian rotations as separate columns, based on the PaleoMAP model[80].

## Spatial clusters as regions

Spatial clusters of sampling, which we termed regions elsewhere in the manuscript, were expected to be more similar in mean temperature and species composition within than among regions per time zone. The species recorded in each of these regions per time zone then became the assemblage of interest (analogous to quantification of thermal bias for observed assemblage over a sampling transect in ref. [11]). Collections were pooled into unique spatial coordinates per time zone. Objective and non-overlapping regions were identified using hierarchical clustering of Euclidean distance matrices of occurrence paleocoordinates of all time zones pooled. We expected these clusters to arise mainly from sampling patterns, given that clustering so far used no palaeocological data. However, regional assemblages should be ecologically distinct, having more differences among them than within them. To assess palaeocological similarity among the clusters defined by Euclidean distance of coordinates, we also estimated groupings of late Pliensbachian occurrences by hierarchical clustering of Jaccard distance matrices of species presences and absences (i.e. using palaeocological co-occurrence but ignoring spatial coordinates). Jaccard distance clusters with less than 14 species were removed to balance the tendency of small samples to drive dissimilarity (via species absences) against persistent and more relevant, larger groupings. Palaeocological clusters validated the use of the separately identified spatial clusters as distinct species assemblages, such as from separate bodies of water or habitat. Adopting ten spatial clusters maximized the agreement between the two approaches.

Finally, practical requirements for spatial clusters included (1) being sampled in different time steps, ideally throughout, and (2) having sufficient occurrences ($n > 25$ per time step). This was the case for four of the eight spatial clusters: the northern and most likely terrestrially influenced British basins cluster, and three clusters surrounding the landmass of Iberia: to the west, to the north, and to the east (likely to be the most pelagic influenced cluster). The benthic fauna of a fifth, Germanic basins cluster were well-sampled in the late Pliensbachian, but not in the early Toarcian. However, its outcrops are exposed throughout our temporal focus, suggesting that species absences were driven by anoxic bottom waters[53] rather than by poor sampling, so this cluster was also used for analysis. Regions derived from these clusters had different thermal regimes (see Results) and variables like terrestrial influence (see Discussion).

## Rates of species occupancy responses

Comparing the sampling-corrected relative rates of extinction or origination rather than their raw proportions made the largest difference to the observed patterns of extinction and origination (Supplementary Fig. 9). Uncorrected extinction rates were highest in the Spinatum but when corrected show not much difference between Spinatum and Tenuicostatum. Uncorrected origination rates were highest in the Spinatum then declined. However, when rates were corrected, origination was much higher in Tenuicostatum. The main conclusion is that correcting observed proportions into more comparable rates, that account for uneven sampling over time, makes a large difference to the results. While several of the correction approaches shown in Supplementary Fig. 9 are more complicated, with no straightforward way to apply them to our goals, the three-timer approach of Alroy[85] is easily adapted as a precaution against spurious features of sampling patterns. Essentially, this focusses on the occupancy response rates in the better sampled taxa. Sampling completeness was relatively high throughout the study interval but highest in the Tenuicostatum, making the patterns centred around this ammonite zone most reliable to interpret.

As a precaution against spurious features of sampling patterns such as those highlighted in the previous paragraph, we focus on comparing numbers of regional two-timer species, that is, species that were observed in a region for at least two time bins consecutively

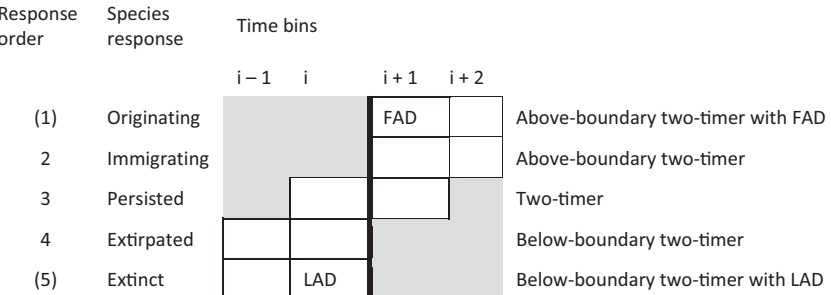

**Fig. 5 | Designation of region-specific species occupancy responses observed around a focal boundary.** The focal boundary is emboldened, with the ammonite zone immediately preceding it being time i. We focus on the responses of regional two-timer species, specifically the lower two-timers for extirpated or extinct species, boundary crossers for persisting species, and upper two-timers for immigrating or originating species. These responses are ordered with respect to expectations of thermal bias. FAD First Appearance Date, LAD Last Appearance Date.

(Fig. 5)[85]. These represent the better-sampled species, whose occupancy responses, their observed changes in regional presence or absence, may be more reliable. The same can be done using three-timers (species must be observed in a region for three time bins consecutively; see below). However, using two-timers has the advantage that the temporal focus of change is a single boundary between two time bins, which fits understanding of the timing of the climatic changes investigated here, rather than more diffuse change over a central bin and both of its demarcating boundaries for three-timers. The well-sampled nature of two-timers and high sampling completeness of the focal ammonite (sub)zones of European regions for this interval means the observed times of extinction, extirpation, immigration or origination are relatively reliable (e.g. against the Signor-Lipps effect, where the true first and last individuals of a taxon are unlikely to be observed as fossils).

Focusing on region two-timers (Fig. 5), immigrating species were those observed in the region in time i + 1 AND time i + 2 but not in i. Originating species were the same but also had their dataset-wide First Appearance Date (FAD) in time i + 1. Extirpated species were those similarly observed in the region in time i AND time i − 1 but not in i + 1, with those having time i as their Last Appearance Date (LAD) were classed as going extinct. Persisting species were observed in the cluster in times i AND time i + 1. There were fewer occurrences before the first time bin (i.e. in the Davoei Zone, which preceded Margaritatus) and after the last bin (in the Variabilis Zone, which followed Bifrons), limiting the quantity of region two-timer species, so their two-timers were simply required to have a presence in times i − 1 and i + 2, respectively, regardless of region. Species still needed a regional occurrence around the focal boundary, either in time i or i + 1, to be assigned a response category (e.g. extirpated), so this step did not artificially increase numbers of species in any response category, but simply allowed more species to pass the sampling threshold (i.e. meeting definitions in Fig. 5) in the Margaritatus and Bifrons zones. Note that in all cases, due to incomplete sampling, extirpation and immigration are probabilistic events rather than definite.

Two-timer species classified as not going extinct nor originating must have occurrences in the future and past, respectively, of time i, such that their species thermal niche (see next section) is averaged over past and future distributions. Meanwhile, the thermal niches of extinct and originating species were inherently limited to only past or only future distributions, respectively. To address the potential criticism of extinct and originating species having a fixed thermal niche, we focus our analysis on extirpation, immigration and persistence responses, and only secondarily including extinctions and originations.

In the alternative three-timer approach (Supplementary Fig. 10), immigrating species were those observed in the region in time i AND time i + 1 but not in i − 1. Originating species were the same but also had their dataset-wide First Appearance Date (FAD) in time i. Extirpated

species were those similarly observed in the region in time i AND time i − 1 but not in i + 1, with those having time i as their Last Appearance Date (LAD) classed as species going extinct. Persisting species were observed in the region in times i AND time i + 1 AND time i − 1. This means a three-bin window on climate is required to set the context of these responses, with immigrations and extirpations likely occurring asynchronously and possibly driven by different events.

### Analysing species and assemblage temporal change
Analyses were separated between species and assemblage levels. A species' thermal bias was defined as the difference between the regional median temperature for a time zone and the species' thermal median (temperatures averaged over all zone-level occurrences of the species from the Margaritatus to Bifrons zones, the complete interval when occurrences were matched to temperature maps). An assemblage thermal bias, often assumed to indicate net vulnerability, was thus the difference between the median of the constituent species' thermal medians[11,28] and the regional median temperature for a time zone.

We expected a gradient of occupancy responses relative to thermal bias in a warming scenario (Fig. 5), with extinct and extirpated species at one extreme having the most negative thermal biases, originating or immigrating species at the other extreme having relatively positive thermal biases, and persisting species having intermediate thermal biases. Therefore, species-level regressions used species occupancy response as an ordered, continuous dependent variable and species thermal bias as a continuous independent variable, *Occupancy_response ~ Thermal_bias*. Mixed effects accounted for the nested analysis structure, where necessary, with species nested within regions and regions nested within time zones (i.e. a single species can have one response and thermal bias per region per time zone, a single region can be observed in multiple time zones). Being at the extremes of the regression line, species originating or going extinct also have a stronger effect on the regression slope than persisting, extirpated (but surviving) or immigrating species (with past occurrences).

For assemblage-level analyses, we recorded the percentage of a current assemblage that was categorized at the species occupancy response levels of persisting, extirpated, or extinct. We also recorded the percentage of a new assemblage that was categorized as immigrating or originating. The turnover of the current assemblage into the new assemblage (i.e. from time i to i + 1) was also quantified by Jaccard distance. These were each used separately as dependent variables. Independent variables were assemblage thermal bias, regional temperature change magnitude, or the difference in occurrence proportions of occurrences from carbonate or offshore substrates (the most frequent substrate types). Here, mixed effects models nested regions within time zones, but had a low sample size (5 regions × 5 time zones = 25 assemblage data points maximum) and thus a weaker

potential for inference. These regressions were applied in an exploratory framework akin to a correlation matrix to weigh evidence for further research. Models using assemblage thermal bias as an independent variable were inverse weighted for the standard deviation of species' thermal bias. We chose to apply these model expectations for regional species responses at a modern-relevant level of warming (+3 °C). All analyses were performed in R[86] with packages divDyn v0.8.2, corrplot v0.92 to present assemblage-level analyses, nlme v3.1–162 for mixed effects models, and vegan v2.6-4 for clustering[82,87,88].

## Reporting summary

Further information on research design is available in the Nature Portfolio Reporting Summary linked to this article.

## Data availability

The data used in this study came from the Paleobiology Database (https://paleobiodb.org/). Both raw and processed datasets, and code to generate the main results, are available in the Zenodo repository https://doi.org/10.5281/zenodo.14626268. Source data underlying all figures are also provided as a Source Data file. Source data are provided with this paper.

## Code availability

The data used in this study came from the Paleobiology Database (https://paleobiodb.org/). Both raw and processed datasets, and code to generate the main results, are available in the Zenodo repository https://doi.org/10.5281/zenodo.14626268.

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

## Acknowledgements

We are grateful for insightful discussions with Paulina S. Nätscher, Erin E. Saupe, Ádám T. Kocsis, Wolfgang Kiessling. The authors gratefully acknowledge the European Regional Development Fund (ERDF), the German Federal Ministry of Education and Research and the Land Brandenburg for supporting this project by providing resources on the high-performance computer system at the Potsdam Institute for Climate Impact Research. All references that contributed data for this study via the PaleoDB are listed in a secondary bibliography in Supplementary Data 3 and we further thank PaleoDB data enterers and authorizers. CJR and MA were supported by German Research Foundation grant number AB 109/11-1. This work is embedded in the Research Unit TERSANE (German Research Foundation grant no. FOR 2332: Temperature-related stressors as a unifying principle in ancient extinctions). This is Paleobiology Database publication number 515.

## Author contributions
M.A. and C.J.R. conceived the project and compiled the occurrence data. M.A. and C.V.U. evaluated the regional climate changes. M.A. vetted the occurrence data. J.P.L and G.F. ran the climate models and extracted data from them. C.J.R. analysed the data and plotted the figures. C.J.R wrote and revised the manuscript with important input from M.A., J.P.L, G.H.M., C.V.U., and G.F.

## Funding

## Competing interests
The authors declare no competing interests.
