## [Peer Review file · Nature Communications]

Marine species and assemblage change foreshadowed by their thermal bias over Early Jurassic warming

Corresponding Author: Dr Carl Reddin

Version 0:

Reviewer comments:

Reviewer #1

(Remarks to the Author)
Please see attached PDF.

(Remarks on code availability)
I ran the R code. I successfully generated Figs. 2 and 3 but obtained the following error (and others, all resulting from this first one).

```
> coastlines <- fetch(dat="paleomap",var="paleocoastlines")
Error in fetch(dat = "paleomap", var = "paleocoastlines") :
Invalid 'src' argument.
> plimargin <- coastlines["185",1]; plic Coast <- coastlines["185",2] #185Ma
Error: object 'coastlines' not found
> toamargin <- coastlines["180",1]; toacoast <- coastlines["180",2] #180Ma
Error: object 'coastlines' not found
```

Reviewer #2

(Remarks to the Author)
This study seeks to understand whether the thermal tolerance of species predicted their origination, immigration, and extinction patterns in the Early Jurassic. The study focuses on well-sampled European sites from the late Pliensbachian through the middle Toarcian. In general, I admit that I find these sorts of studies difficult to evaluate. On the one hand, identifying overarching ecological trends across extinctions is increasingly important; however, it is also difficult to perform a study at this scale with the level of detail that is critical in (paleo)ecological studies. Please note that I could not access the code nor the supplemental dataset for this paper, so my review was limited to the main text.

My main concern is the combination of a circular argument and the over-generalization of various aspects of the study. In my opinion, these must be addressed (or at least acknowledged) before the article can be accepted. Specifically:

- 1) I appreciate the difficulty in working with such a large dataset, but to define the species thermal median as the "temperatures averaged over all zone-level occurrences of the species from the Margaritatus to Bifrons zones" will, by definition, result in a cold preference for all species that go extinct and a warm bias for all species originating. Surely one would get a more accurate thermal median by looking across the entire range the species existed? As it is currently set up, the results seem strongly affected by confirmation bias (not only with the temperatures being limited to this time interval, but also that other environmental controls were not, or were minimally, evaluated).
- 2) I am not comfortable with the generalization of a niche as simply thermal tolerance, especially with the access to PaleoEcological Niche Models. To primarily/only look at thermal sensitivity and then conclude that it has the greatest impact, is again an issue of confirmation bias.

Generalization issues also come up in several other places:

2a) In many places the authors apply their findings to all marine taxa (or at least that is implied by how the text is written, especially in the Discussion), but the dataset is limited to 3 groups (which is fine, those are well sampled), so I suggest

rephrasing some of the more "overarching" conclusions (i.e., this study is not necessarily predictive of how ALL marine communities will respond).

2b) I am not sure why a linear model was used when the trends in the data do not appear to be linear (Fig. 2D, 2E). I know a paper was cited to support the linear model (lines 264-266), but I think other models should at least be attempted in some cases. For example, although the R² value shows significance, it seems like ALMOST ALL taxa reported in D are "cold biased" with significant scatter in the persisting and extirpated taxa.

2c) As a follow-up on 2a&b, expanding the predictions to +3C seems unfounded (e.g., Lines 228-230, Fig. 4), especially considering the limited time window studied and the poor fit of some of the models. Again, the argument feels circular to me, especially given the limited focus of the study (i.e., bivalves, brachiopods, gastropods, Early Jurassic, Europe) as well as the variable responses amongst taxa. This also assumes that species' niches and thermal tolerances cannot evolve, which should be directly stated in the text.

3) In Ecological Niche Modeling (ENM), one of the important assumptions is that niches are species-specific, rather than at an assemblage or family level. I am not suggesting the authors toss out the entire study, but it seems like an ENM would have been a better choice for a robust analysis of this type (or at least would back up the assumption that thermal tolerance is the main driver of diversity patterns). In that same vein, it seems like the analyses of depth and substrate were an afterthought, although I could not access the supplemental data, so maybe these issues were explored sufficiently there.

Other issues:

- Several acronyms were not defined (CIs, Cis), and explaining some jargon would make the paper more accessible (e.g., I am always surprised how few people are familiar with extirpation).
- Some sentences are awkward or I had a hard time understanding what the authors were saying (e.g., lines 64-66, 77-79, 126-129, 140-142, 222-227)
- A few minor grammatical issues (e.g., lines 74, 475, 600)
- What are the numbers in Fig. 1A?
- I would have liked to see the climate data from each of the ammonite subzones. Maybe this was in supplemental.
- How was a "change in habitat substrate" determined (line 142)? There is a lot of discussion about habitat, but no explanation of how it was determined; again, maybe this was in supplemental data.
- Lines 147-151: Please show the results for the bivalves/brachiopods as well as all the data together. Without the data the claim is not substantiated.
- Line 201 needs a citation, again, niches should be assigned to species.
- What do you mean by "the share" of a group (e.g., 237)? Please explain.
- Line 253: "Escalating response gradient" This conclusion is not always founded based on your dataset.
- Lines 309-313: What data is this referring to? The results in Fig. 2 seem to disagree in places (especially concerning the immigrating/originating species).
- Lines 415-417: This conclusion is not shown in the dataset, or at least I am sufficiently confused about when STI vs. CTI was used.
- Late, Middle, and Early Pliensbachian/Toarcian should not be capitalized as only the stages are official (i.e., it should be late Pliensbachian, or middle Toarcian, etc.)
- Line 542: define "sufficient occurrences".

(Remarks on code availability)

The dataset was restricted to "users with access" so I could not view it.

Reviewer #3

(Remarks to the Author)

I was excited to review this paper, but I found it disappointing. The title does not accurately convey the contents of the work. The analysis considers a limited geographic region and taxa groups. The reasons for the selections of region, taxonomic groups and coarse temporal resolution used for the analyses are not clear to me.

The paper is written as though an important general empirical model is proposed, but it is based on data from three taxa groups in one small region of the world over one event. I did not find the results particularly interesting or useful, and they needed better embedding into the literature in terms of what we know about present and future climate change, or that occurring in the Toarcian.

Despite this being a really well-sampled region for the event, data are combined into coarse spatial and temporal units. The amount of change explained is modest at best (18%) and this is based on only a subset of the data used. Interpretations are also based on non-significant statistical results.

Insufficient consideration is given to the importance of other environmental drivers, and the considerable influence of biotic factors on structuring populations, communities and ecosystems across broad spatial scales. The effects of species interactions seem to be almost entirely neglected, but they can be the primary driving factor (e.g. classic experimental marine ecology throughout the 70s, 80s, 90s, 00's).

Almost no consideration seems to be given to uncertainties on species distribution data, taxonomic identification, its stratigraphic constraints and so on. The grouping by space and time is not well explained or reasoned. There is some very highly temporally constrained data available for the TOAE, but it is lumped into ammonite zones most of which are hundreds

of thousands of years duration.

If this paper is supposed to be written for a general audience of Nature journal readers it is not effective, e.g. a non-geological audience would not know what an ammonite zone is or means, nor would they appreciate first and last appearance, or what the Signor-Lipps effect is. None of the concepts or approaches are referenced, e.g. those that are conceptual/artefactual (Signor-Lipps), analytical (Euclidean distances, cluster analyses).

It lacks the polish or clarity I expect for a submission to this journal. The style of expression, choice of language, explanation of the approaches chosen and the assumptions made will not engage a broad audience.

I found the abstract and introduction vague and not always well-supported by the literature. Understanding of changes in species biogeography is poor, and over simplified.

Introduction:

Overall, the literature cited is insufficient and is missing key contributions. Many key points are made citing no references, or using a single reference – this does not demonstrate a robust evidence base exists nor incorporate differing perspectives.

L37 In general, BUT there are species that have expanded ranges, contracted ranges, shifted ranges, some that have deepened their distribution, and some that have not changed at all.

L38 Vague. the amount matters if you want to make comparisons modern and ancient

L42 Meaning of “performance decline”?

L47 more needed

L52 unclear

L54 linked to species latitudinal shifts

L57 Hypoxia and anoxia

L57 other toarcian refs needed to support this - but also from other time periods, and modern

L58 inferred/determined from? i think thats what you've done?

L60 is optima the important component? or is maxima and minima that has deleterious effect, plenty of species occupy a realised niche not a fundamental niche

L64 Thermal bias: the concept needs exploring more. I would expect species distributions to be defined by their thermal limits, i.e. the range of temperatures not the optimal. In either case where is this key assumption/interpretation explained?

L69 “the wider validity is rarely tested” so how can it be used with confidence to make meaningful interpretations for fossils?

L87 unclear

L74 “Warm” is vague

L76 “persisting species” needs defining

L90 needs defining

L91 Why only these taxonomic groups?

L102 what literature?

L104 how do you derive a summer temperature from geological/palaeontological materials?

Was the analysis completed on species or genera? How were taxa classified as immigrating etc? Is the analysis on assemblages or species – what are the assemblages composed of?

How were thermal niches determined, what are the assumptions?

Many assumptions are made in relation to the approach that are not explained, i.e. that biogeographic distribution reflects physiological ideals and limits I guess? There is very little explanation of the biological or ecological basis.

The temporal resolution is very coarse. Some of these ammonite zones are thought to be of considerable duration– how does it match current day shifts?

Results

Regular use of “bins” is not helpful, why not just call them categories?

Makes a lot of reference to the supplement – should it really be a supplement?

What is $p < 0.2$ taken to indicate, this is not a significance value used by any statistical analyses I have ever used. If it is to be used then it needs to be discussed and backed up by some discussion and evidence that it could be considered statistically meaningful.

L201 “Faunal responses to climate change are often measured or projected at the level of assemblage.” Not in my

experience, but you should cite your sources so that we can see what works you are referring to.

L221, 224 Why is interpretation made from analyses that are not statistically significant? Similarly non-significant trendlines are plotted on figures 2b, 2e, 2f, 3

Fig. 1, surely temp change is more important than absolute temperature?
I did not find fig. 4 intuitive or easy to read
Figure captions and labelling are unclear – see annotations on the text

The phases of warming, cooling and temporal stages etc are not sufficiently contextualised

Modern geographic references seem to be given priority over palaeogeography in the descriptions which makes no sense given the movement of species would not have been determined by today's geography. E.g. the Germanic basins, the British basins and so on.

How have species absences been dealt with, how do you know a species is missing? There is much mention of varying sample size but how does that reflect the effort taken (or not) to establish that a species was absent? What about stratigraphic or preservational biases? Some of the gastropods in these sections are very small.

Discussion

L249 "removal" meaning?

L250 regional species loss = extirpation, but also insufficient use of literature, it is extensive

L256 taxonomic membership?

L260 Why?

I found some of the interpretations inflated, e.g. line 252 "We present empirical evidence from the fossil record that immigration, persistence, extirpation, and likely the extinction of species form an escalating response gradient linked to species suitability to regional conditions, as estimated by their thermal bias." Based on information on local sea surface temperature only, information that is inferred from proxies? The sources of data used to infer temperatures or thermal bias were not described or critiqued, authors are referred to the supplement for this but surely this is critical?

If thermal bias or temperature tolerances are gleaned only from the fossil distributions themselves, how do we know that their distributions are explained only by water temperature?

L273 There is evidence for very large shifts in body size of up to 50% in the as a response to environmental change in the taxonomic groups, considered analysed in this study that are not cited. It is not simply a case of taxonomic groups of smaller size and life history, there is actually evidence from this event growth was stunted by some factor.

L267 dispersal, so why haven't these factors been considered in this work?

How is the connectivity of ocean basins, or species etc that might have facilitated changes in species biogeographic ranges (or not) considered? Sessile invertebrates have differing living habits and life history mechanisms, some are local dispersers only.

Some basins were restricted and were not always well connected to other regions because of other factors of palaeogeography, habitat loss, food supply etc. Insufficient consideration is given to other influences on their recorded or likely predicted future distributions. Likewise, the success or failure of any changes in species distributions and possible extinctions are not determined only by temperature, biotic factors such as predation and competition are likely to be incredibly important, yet aren't considered at all. Again, any estimates of extinction vulnerability (L277) are only due to temperature.

L277-278 what do you consider to be "rapid warming"?

L284 like the habitat loss that would have occurred during this event where anoxia and ocean acidification were key?

L286 There is good evidence for range shifts in marine invertebrates (these are vulnerable to warming) and some fish are not changing ranges, some are simply expanding and some aren't moving or are going to greater depth. E.g. see Poloczanska et al. 2013

L301 reference needed

There is mention that L320 "Poor dispersal capabilities and/or dispersal barriers can lead to a species' failure to lessen its population thermal biases by shifting distributions, thereby shrinking its geographical range 48, and making it vulnerable to global extinction 45,49." So, why haven't these factors been considered in this work?

L322 "Mechanisms of and limitations to habitat tracking should be explored during other intervals with changes in climate, sea level, and geography e.g. 49." - Why not consider them in this study?

L327, 333, L380 unclear
L338 misses key literature
L390 explain

Citation style is quite unhelpful and uninformative, e.g. L380 “despite evidence often to the contrary.” Followed by no explanation of the contradiction. “Temporal resolution is not a problem per se for the application of paleontological insights to modern issues” followed by no explanation, if you are going to comment on this (which you absolutely should) at least explain why. Nature readers are diverse, you should be reaching a broad audience, especially if you want to reach any audience beyond geologists.

L393 cite evidence for the “current biodiversity crisis”

L411: How did you consider the rare component ?
L413 Of gastropods, bivalves and brachiopods

Approach and methodology:

The methods are poorly presented with much methodological information missing. I have annotated a pdf with detailed comments in relation to this.

I didn't find the further detail in the supplement helpful in addressing my methodological queries

L433, 455, 458, 547 references?

What is the source of the approach on thermal bias?

Information is delivered in the wrong order, the basics of the data collation and analysis are not explained. How is information grouped prior to analyses and on what basis palaeontological/ecological/environmental is not clearly explained (if at all!). Key terms are not defined: what are you considering an “immigrant assemblage” to be, what is a persistent assemblage? How are spatial assemblages defined – I assume this is on the basis of assumed physiological tolerances of those taxa in life. Is the analysis conducted on species? The ecological unit from which a distinct biogeographical range (and by inference physiological limits) might be inferred or is it based on genera? In the methods it says “occurrences initially had to be accepted at least at the genus level”, but there is no follow-up statement that I can see.

Writing style:

In many places the text is impenetrable to anyone other than the authors. Punctuation is poor. Sentences are long and technical (often 4-5 lines in length),
Definitions (even the authors own that are introduced for just this work) are lacking. There is no explanation or discussion of the ecological meaning of some of them and the core assumptions on which the analysis is based.

Some style of writing overly mechanistic, clumsy and non-intuitive, e.g. “Non-temperature habitat (and sampling) heterogeneity is likely to dominate at scales beneath our effective spatial resolution” what is “non-temperature” habitat?

The term ecology is frequently used in place of the correct term palaeoecology for much of the contents of the paper.

(Remarks on code availability)

Version 1:

Reviewer comments:

Reviewer #1

(Remarks to the Author)

I did enjoy reading this MS by Reddin et al. again. I still believe this is a very valuable contribution representing an impressive amount of work. The MS remains relatively technical, but this is probably necessary to correctly convey the results of these complex analyses and the authors did great in revising the MS to make it much more accessible. I also appreciated the extensive supplementary information. I think the MS is now suitable for publication, which I definitely support.

My only concern regards the code hosted on Zenodo, which I suggest to update before publication (see dedicated section below).

Best regards,
– Alexandre Pohl

(Remarks on code availability)

I ran the code, and it now runs well, but:

(1) figures are not saved on disk, which is not very convenient

(2) figures do not seem to perfectly match the revised MS. For instance, Fig. 4 is the original version, not the revised one (heatmap). The README also refers to the previous title.

I encourage the authors to update this Zenodo repository before publication of the MS.

Reviewer #4

(Remarks to the Author)

Dear authors, I am serving as a fourth reviewer assessing your manuscript. I see how your results are relevant in the paleontological context and, more importantly, in marine conservation. The STI/CTI approach has become extremely popular among conservation biologists, but evidence of its value in anticipating the future of biodiversity needs to be more extensive. Your study is a significant step forward. My only criticism is aligned with the second comment of reviewer #2: STI values should be estimated for the species' entire geographic distribution (Burrows et al. 2019, Supplementary Table 2). I understand the difficulties explained by the authors in getting these values, but this should be stated as a potential shortcoming of the analyses. Other than that, I enjoyed your manuscript.

Burrows, M.T., Bates, A.E., Costello, M.J., Edwards, M., Edgar, G.J., Fox, C.J., Halpern, B.S., Hiddink, J.G., Pinsky, M.L., Batt, R.D. and García Molinos, J., 2019. Ocean community warming responses explained by thermal affinities and temperature gradients. *Nature Climate Change*, 9(12), pp.959-963.

(Remarks on code availability)

Reviewer #1 (Remarks to the Author):

Paper summary:

Reddin et al. combine global climatic simulations and paleontological data to investigate the relationship between species thermal optima and their response to warming during the Early Jurassic (including the Toarcian OAE interval) on the European Tethys shelf. Paleontological data consist in a curated PBDB subset. Paleoclimate data correspond to CLIMBER-X results bilinearly interpolated to the resolution of HadCM3. The authors estimate the thermal preferences of brachiopods, bivalves and gastropods by extracting simulated ocean temperatures at their (paleo-) spatial-temporal locations, and then determine for each of the 1-Myr time slices considered, the difference between the thermal preference of the species and the temperature experienced by the species during the time slice. They show that the magnitude of this difference (or ‘thermal bias’) generally constitutes a good predictor of the response of the species to warming: origination, immigration, persistence, extirpation and extinction. They show that their conclusions stand when origination and extinction are excluded from the analysis and when the climatic scenario is varied in the model. Finally, the authors provide an extensive discussion of the limitations and implications of their results, notably in the context of the ongoing global climate change.

General comment:

I think that this manuscript overall constitutes a very interesting and robust work and would be a valuable addition to the field. It would be of interest to scientists from different communities, ranging from paleontologists to (paleo)ecologists and modern-day biologists. Therefore, I encourage its publication in *Nature Communications*. However, I think that the manuscript requires significant revisions to make it accessible. While the introduction, the discussion and the methods are well written, I think that the results are too difficult to follow and encourage the authors to significantly revise that part to make it accessible to a wide audience. I think that this is really a necessary step before considering the publication of this manuscript.

#We sincerely thank the reviewer for their time and very useful thoughts, which have resulted in greater written clarity, revised main figures, and additional supplementary figures.

Main comments:

1. Title and wording.

- I only understood the notion of escalation relatively far into the manuscript, and thus think that it may help removing this concept from the title and abstract.
- In general, I think that the wording should be revised to be clearer. An example:
 - Lines 28–29: “the relationship was overridden by severe seawater deoxygenation”: I am not sure this is grammatically correct and think it is difficult to understand this sentence before reading the main text.

#We thank all three reviewers for pushing us to improve the clarity, which we think is now improved. We have now replaced the word ‘escalation’ with either ‘sequence’ or ‘gradient’ throughout the manuscript and supplementary materials, and the title has been rewritten. The comments of all three reviewers have also been very helpful to direct where the manuscript most needed clarification.

2. Missing parts. I could not find the (apparently extensive) supplementary information. I was also unable to access the code, since the data hosted on Zenodo is not publicly accessible. In particular, I would be interested in seeing maps showing the patterns described on lines 159–162: the northern part of the Tethys is the highest in latitude and thus the coldest – ok. I have difficulties understanding how these regions may undergo the smallest temperature increase during the first

warming phase: I would encourage the authors to check, and explain the mechanism behind this unexpected (hence potentially interesting) response.

#As corresponding author, I sincerely apologise for forgetting to upload the supplementary information. Indeed, this led to unnecessary confusion for all reviewers. We now add the maps of Fig. R1 to the SM as the new Fig. S4.

We thank the reviewer for urging us to investigate how the northern regions might undergo the smallest temperature change during the first warming and the largest during the second, as we previously observed. After doing so, the northern regions appear to *always* experience the largest degrees of warming. However, the difference in relative temperature changes over the two climate phases, which we previously described, is largely an artefact of using the temperatures at fossil occurrences to indicate environmental change. Therefore, although we describe below how the temperature change at occurrences and the temperature change based on the climate model output rasters differ, we opted not to include most of it in the manuscript as it distracts from the storyline. Note that this artefact is only relevant where occurrence paleocoordinates are used to estimate the environmental temperatures, and does not affect thermal bias estimation because in that case the temperatures at the occurrence paleocoordinates are exactly what we are interested in (inhabited temperatures).

Below are the 'warming differences' plots (Fig. R1), with the top row being the before-after and difference plots for the first warming phase (note at this temporal scale, this ignores the very rapid warming at the Pliensbachian-Toarcian boundary), and the bottom row being the before-after and difference plots for the second warming phase (into the T-OAE). During both warming phases (and at the boundary), the largest warming is observed at the more northern regions. However, such coastal regions are also where the global model may be weakest as factors like tidal mixing become more important.

The temperature changes at occurrences (Fig. R2) show the large jump in temperature during the second warming phase, which appears particularly marked at the northern regions, as we previously described in the main text. However, the paleolatitudinal change at occurrences (Fig. R3) shows how the occurrence sampling at the northern regions 'compensates' for the first warming phase by shifting polewards (=cooler, all else being equal) but then 'synergises' with the second warming phase by shifting equatorwards (=warmer, all else being equal). Occurrence paleolatitudes at the other regions do not show the same paleolatitude change pattern.

We opted to change the regional temperature (change) values in the main text to cell medians, rather than the sampling-influenced at-occurrence temperature medians.

Fig. R1. Comparing geographical patterns of SST over the two warming phases. Panels A, B, D and E use the same colour scale, highlighting how cooler habitats disappeared from the region over time. Panels C and F use the same colour scale, highlighting how mean warming may have been greatest in northern regions (note that panel C does not capture rapid warming over the stage boundary at this temporal resolution). These (sub)zones cover the range of temperature values of the main CO₂ scenario, from Spinatum at 400 ppm to Exaratum at 1000 ppm (see Table 2 and global maps, Fig. S8, for additional scenarios). For comparison with Fig. 1, these show the Toarcian paleogeography (i.e. maximum sea-level coastlines for the study interval). The coverage of occurrences per (sub)zone is shown (A, B, D, E only) using the same colouration as Fig. 1.

Fig. R2. Boxplots of modelled seawater temperatures (SST) at regional sampled occurrences, rather than over the entire coverage of the region. This highlights how sampling can exacerbate or compensate for modelled changes in regional temperature, resulting in observed changes slightly different to the original modelled changes (via CLIMBER-X). Panels are the region names plotted approximately according to paleocoordinates, with N, E and W being abbreviations of north, east and west. X-axis labels are abbreviations of the ammonite (sub)zones, through Margaritatus, Spinatum, Tenuicostatum, Exaratum, Falciferum, and Bifrons.

Fig. R3. Boxplots of regional sampled occurrences paleolatitudes (plat) in the Toarcian (Toa). Assuming higher latitudes are cooler and lower latitudes are warmer, this highlights how sampling can exacerbate or compensate for modelled changes in regional temperature. Some of the observed temperature changes in Fig. R2 are likely because of changes in sampling paleolatitude, rather than changes in modelled regional temperature. See Fig. R2 for further details.

3. Suggestions to make the results clearer.

- I would suggest to avoid any code-like language and shortcuts and instead expand on these notions in the form of textual explanations; e.g.:

- Line 73: "(regression: response~thermal_bias)". Such phrasing does not make it easier (or pleasant) to the reader to understand what the authors mean.

- Line 80: "(hypothesis A regression slope becomes steeper)"

#Both of the above parts are now removed and text has been revised to improve clarity.

- What is R_{marginal} on line 139? I do realize my background in statistics is quite weak, but still think that the text should be assessable to a wide audience.

#The text has been amended so it is understandable without understanding this metric. Marginal R-squared is the proportion of total variance explained by the fixed effects of a model alone, which are usually the variables of interest, without the proportion explained by random effects.

- Table 1, in general, is very difficult to understand. Line contents are relatively obscure and column headers are not defined.

#We now clarify the meaning, definitions, and add interpretative support for this regression model ANOVA table, both in the caption and in the main text. At the editors discretion, we could move this table to the SM and move the most important values to the main text, but we think it is most efficient to see the multiple regression model as one in the Table here. The use of interaction terms in this model would require a lot of space if it was translated to figures.

○ The same for Fig. 3: the label of the y axis is difficult to understand and not explained.

#This figure has now been simplified by rephrasing the y-axis to 'mean difference in thermal bias between response levels, °C', and removing the extra lines (the previous version with the extra lines, with an updated y-axis, is now in the supplementary materials, Fig. S2A).

○ Please define the 'mixed-effect model' (and ensure a consistent spelling throughout, "effect" vs. "effects").

#First mentioning now changed to show that a mixed effects model, now consistently spelled throughout, incorporates both fixed effects (typically those we are interested in) and random effects (those effects that relate the sampling structure to the wider sampling population).

• Possibly include a summary of key explanations that are only found in the Methods, such as the way that thermal bias is calculated and how the clusters/regions are used in the analyses. I did get a much clearer vision of what was done in this study after reading the Methods.

#Thanks, the thermal bias definition has been moved to the hypothesis paragraph of the Introduction (second to last paragraph therein) and the nesting structure, including the handling of regions, has been expanded here too.

• In general, it would be good to better introduce the paragraphs and ensure better transitions. For instance, the authors may want to emphasize that they first work at the species level, and then look at assemblages. In particular, the caption of Fig. 3 suggests that this analysis already considers assemblages?

#To avoid this confusion, we have now removed any reference to 'assemblage' in the results before its dedicated section, 'Assemblage-level thermal bias and responses', including from the caption of Fig. 3.

• I wonder whether Fig. 4 may not benefit from a major simplification, converted to a simple heatmap.

#We have adopted this suggestion to use a simpler heatmap. We added the numbers in %, which are referred to in the main text, and have simplified the figure caption for the general reader.

4. Suggestions of citations. These are really just suggestions since I don't especially want to push citing my own work, but I feel that these 2 papers may fall exactly within the topic and thus provide the references here in case it would help:

• When discussion the potential, combined impact of temperature and ocean oxygenation, the authors might be interested in referring to this recent study (<https://www.science.org/doi/full/10.1126/sciadv.adg7679>) where we provide quantitative analyses of the impact of these environmental variables on extirpation and extinction rates simulated in response to global warming.

#Many thanks. This reference exemplifies some of our recommendations, such as integrating physical (tectonic) and biochemical (ecophysiotype) modelling, so we have added it in a couple of instances in the Discussion.

• Here (<https://agupubs.onlinelibrary.wiley.com/doi/full/10.1029/2018PA003394>) we compiled geological evidence for the north-south redox gradient in the Tethys and used general circulation models to try and provide explanations, notably focusing on the notion of salinity-driven stratification along the northern coast.

#Thank you for bringing this very useful one to our attention (also for the future). We have added it to several places in the Discussion, focussed around current directions, salinity differences and stratification, and how this likely encouraged anoxia in the northern shelf.

Minor and technical comments:

- Fig 1: “E., W. and N. Ibera are east, west and north of Iberia”. Here and in the methods, the authors refer to such clusters, but they are not shown on the map.

#The region names are now placed directly on the map rather than on the legend.

- Fig. 2: would it be possible to show the interval of T-OAE?

#This is not straight-forward as the x-axis is not time-continuous but time bins, and the T-OAE interval is beneath this temporal resolution. We have now noted in the caption that “The T-OAE occupies most of the Exaratum subzone”.

- Line 256: what does “taxonomic membership” mean here?

#Changed to “the species’ taxonomic membership grouping”.

- The authors used bilinear remapping to interpolate their climate model output. Isn’t this method inducing biases, notably creating temperature values that are not present in the original model? Wouldn’t be the near-neighbor method more conservative?

#Below we illustrate the differences between the mapped outputs at two CO₂ concentrations (Figs. R4, R5), showing (A) the original raster at the CLIMBER-X resolution, (B) the data ‘downscaled’ by nearest neighbour values, and (C) the data downscaled by bilinear interpolation. B and C (middle and lower panels per figure) are the same resolution as the HadCM3 outputs, but appear the same resolution as the original CLIMBER-X outputs because, we believe, the nearest neighbour is overly conservative, resulting in unnaturally sudden steps in temperature change. Instead, we quantitatively support our downscaling of the CLIMBER-X outputs by (a) showing and quantifying the similarity of the SST features produced by the two models, and (b) not exceeding the resolution of the HadCM3 outputs. Interpolation does produce values not present in the original model output, but we believe we have used an approach that conservatively makes use of the information we have available (especially, multiple outputs from both the CLIMBER-X and HadCM3 models). We have now added some of these details to the methods section, ‘Seawater temperature maps’.

Figure R4. Sea surface temperature estimates (in °C, scale bar on right) of the 400 ppm CO₂ scenario with the Toarcian paleogeography (180 Ma), showing the original CLIMBER-X output (top panel), the data downsampled by nearest neighbour values (middle), and the data downsampled by bilinear interpolation (lower). Downscaling is to the same resolution as the HadCM3 model, which is quantitatively demonstrated to produce similar temperature features as the CLIMBER-X model.

Figure R5. Sea surface temperature estimates (in °C, scale bar on right) of the 1000ppm CO₂ scenario with the Toarcian paleogeography (180 Ma). Further details as in Fig. R4.

- Line 486: I am not sure one can define a “best” estimate of climatic sensitivity, but this value is effectively within the range defined by recent models and not excessively lower than the multimodel mean of CMIP5 and CMIP6

#We agree, and have changed the wording to: “While both models have an equilibrium climate sensitivity well within the range of CMIP6 models, the HadCM3 model is more sensitive than CLIMBER-X and generally yields higher temperatures at elevated CO₂ levels.”

Reviewer #1 (Remarks on code availability):

I ran the R code. I successfully generated Figs. 2 and 3 but obtained the following error (and others, all resulting from this first one).

```
> coastlines <- fetch(dat="paleomap", var="paleocoastlines")
Error in fetch(dat = "paleomap", var = "paleocoastlines") :
Invalid 'src' argument.
```

```
> plimargin <- coastlines["185",1]; plicoast <- coastlines["185",2] #185Ma
Error: object 'coastlines' not found
> toamargin <- coastlines["180",1]; toacoast <- coastlines["180",2] #180Ma
Error: object 'coastlines' not found
```

#Thanks for noting this. We have updated the R code to include the 'coastlines' object in an RData file, which saves the reader having to import this file from an external source (R package 'chronosphere', while the coastline data are from Kocsis & Scotese 2021).

Revisions to figures 3 and 4 have been updated in the code.

Reviewer #2 (Remarks to the Author):

This study seeks to understand whether the thermal tolerance of species predicted their origination, immigration, and extinction patterns in the Early Jurassic. The study focuses on well-sampled European sites from the late Pliensbachian through the middle Toarcian. In general, I admit that I find these sorts of studies difficult to evaluate. On the one hand, identifying overarching ecological trends across extinctions is increasingly important; however, it is also difficult to perform a study at this scale with the level of detail that is critical in (paleo)ecological studies. Please note that I could not access the code nor the supplemental dataset for this paper, so my review was limited to the main text.

#We thank the reviewer for their time and insight, which has substantially helped our manuscript. We respect that these interdisciplinary works, resulting from the input of climate modellers, interval specialists on taxa and geochemical indicators, and statistical modellers, are challenging to review, and we hope our revised version communicates our findings better.

My main concern is the combination of a circular argument and the over-generalization of various aspects of the study. In my opinion, these must be addressed (or at least acknowledged) before the article can be accepted.

Specifically:

1) I appreciate the difficulty in working with such a large dataset, but to define the species thermal median as the "temperatures averaged over all zone-level occurrences of the species from the Margaritatus to Bifrons zones" will, by definition, result in a cold preference for all species that go extinct and a warm bias for all species originating. Surely one would get a more accurate thermal median by looking across the entire range the species existed? As it is currently set up, the results seem strongly affected by confirmation bias (not only with the temperatures being limited to this time interval, but also that other environmental controls were not, or were minimally, evaluated).

#As corresponding author, I again sincerely apologise for forgetting to upload the supplementary information, which addresses some of these points, especially that of

evaluating other environmental controls. This led to unnecessary confusion for all reviewers.

We anticipated the ‘circular argument’ criticism and did our best to address the issue of ‘a cold preference for all species that go extinct and a warm bias for all species originating’ by running the analysis with different treatment of the species going extinct and originating, including leaving them out entirely, which did not change the results (Tables 1, S1 & S2). Those species are the most affected by that criticism. For clarity, we now move the sentence from methods to results, starting its own paragraph: “To guard against criticism that originating and extinct species’ thermal niches were pre-decided (e.g. since species going extinct in time i can only have occurrences in the past relative to time i , when climates tended to be relatively colder in our study), we compare regression results with extinction or origination responses left out vs. included.” We hope to have communicated this validation test better.

The recommendation of ‘looking across the entire range the species existed’ is intuitive but is complicated:

(1) The late Pliensbachian to early Toarcian interval covers a transition from the coolest global temperatures of the Early Jurassic to the warmest Early Jurassic global temperatures (e.g. Dera et al. 2011). Therefore, even if the temporal focus was wider, we wouldn’t encounter warmer or cooler global temperatures and the results would likely be unaffected. We also now add to the late introduction that, “The late Pliensbachian to early Toarcian interval covers a transition from the coolest global temperatures of the Early Jurassic... to stabilisation as a greenhouse climate with the warmest Early Jurassic global temperatures”, supported by the reference of Dera et al. 2011.

(2) Taking a wider temporal focus/ longer temporal extent limits the resolution we can make our analyses at. One of the strengths of our study is that we look at species-level change over regions at substage temporal resolution, while most similar studies (with a longer temporal extent) look at genus-level change at stage resolution. Outside of our focal interval i.e. Davoei Zone, preceding the Margaritatus Zone, and the Variabilis Zone, which followed Bifrons, occurrences are far fewer and some regions are simply unsampled, meaning we are already at the temporal extent limits for our analysis. Furthermore, it becomes less likely that the focal species pool would still exist (either having not yet evolved or having gone extinct). Therefore, a wider focus might require an investigation of genus-level regional occupancy change at stage resolution, which we agree would be interesting, but not what we aimed at here.

To be honest, despite how we as scientists must be alert for circular reasoning, there is ‘circularity’ inherent in many natural systems that involve feedback mechanisms, including how we understand species thermal niches to evolve. Species adapt to (previous) experienced temperatures and use those adaptations to tolerate future temperatures in a given location or to settle into new locations (which thence feeds

back as species success in a location). In other words, evolution uses space-for-time substitution to maximise a species temperature-dependent success. But testing how thermal niches evolve is not the novelty or the point of work such as ours. We expect species to respond in the manner we hypothesised but we do not *know* if our expectations are always supported in the past. This is especially the case, as discussed in this peer review, because species respond to more than just temperature. Maybe there are unknown surprises as warming magnitudes increase and, if there are, we need to find out about them before they occur in the near future. It is in this validation sense our results are useful.

2) I am not comfortable with the generalization of a niche as simply thermal tolerance, especially with the access to PaleoEcological Niche Models. To primarily/only look at thermal sensitivity and then conclude that it has the greatest impact, is again an issue of confirmation bias.

#We very much agree with the reviewer that a species has many more requirements than a suitable temperature range. In our statistics, these other requirements will make up the unexplained variance in the responses we observe. In this manuscript, our aim is not to show that temperature has the greatest impact. Indeed, temperature is not being competed against any alternatives, except at the assemblage level with depth and substrate changes. But we do aim to show that temperature, especially thermal bias, is an important predictor -- not to say it is the only predictor. There are many modern ecology studies that evidence and focus on the importance of temperature on setting a species' distribution (several referenced here e.g. Tittensor et al. 2010). Sunday et al. (2012) show how marine species range edges in particular tend to closely match their thermal tolerances. This serves as the background for the study of STI, CTI and thermal bias (Devictor et al. 2008; Stuart-Smith et al. 2015; Day et al. 2018; Bonachela et al. 2021) and our own study on fossils. However, we do also investigate the effect of substrate (carbonate vs siliciclastic), depth and their changes, and conclude that at this scale they are not as important as temperature (i.e. a region likely has more heterogeneity in depth and substrate type than it does in annual mean temperature, and a species only has to be present once in a region per zone). We have made several edits for clarity based on this comment and make sure we always refer to 'thermal' niche, when that is what we mean (i.e. vs other niche dimensions).

Generalization issues also come up in several other places:

2a) In many places the authors apply their findings to all marine taxa (or at least that is implied by how the text is written, especially in the Discussion), but the dataset is limited to 3 groups (which is fine, those are well sampled), so I suggest rephrasing some of the more "overarching" conclusions (i.e., this study is not necessarily predictive of how ALL marine communities will respond).

#We agree that the conclusions might not be generalisable to all marine taxa, but we cite evidence from modern studies that have quantified the utility of one clade (or

subset of) to indicate the patterns of another group. This supports that our results should be robust for marine benthic invertebrates but we agree that all marine taxa would be too far. We have added the wording to the Introduction:

“Here, most marine benthic fossils are bivalves, brachiopods, or gastropods, whose species-level taxonomy is well-agreed, thus we focus on these three groups. Coastal taxa tend to be congruent in their diversity patterns (Tittensor et al. 2010), and richness patterns of marine molluscs, even limited to their most common species, serve as good indicators for other marine ectotherm clades (Reddin et al. 2015).”

We have also added more reminders that the patterns are based on mid-latitude bivalves, brachiopods, and gastropods (e.g. first Discussion paragraph).

2b) I am not sure why a linear model was used when the trends in the data do not appear to be linear (Fig. 2D, 2E). I know a paper was cited to support the linear model (lines 264-266), but I think other models should at least be attempted in some cases.

#The linear model reflects the *a priori* scientific understanding at hand to form the hypothesis. It also fits our expectations best regarding how occupancy responds to thermal bias e.g. the gradient. Modelling the relationship as linear is also the most simple and conservative approach, with the least assumptions. Additional assumptions are more risky, especially because our understanding of ancient ecosystems, and how regional topology may affect expression of the thermal niches on species distributions, is far from perfect. For instance, a non-linear relationship could be supported purely because of sampling patterns. In our case, we discuss how there is evidence that some non-linearity is forced by the anoxic regions being situated to the north of all samples. We have expanded some of these points in the discussion and think it is best for now to only discuss non-linear possibilities for future studies, as we do.

For example, although the R² value shows significance, it seems like ALMOST ALL taxa reported in D are "cold biased" with significant scatter in the persisting and extirpated taxa.

We now highlight this “cold biased” observation as, “Negative (cool) thermal biases prevailed during warming associated phases (e.g. intercept coefficient in Table 1), especially visible in warming phase 2, when extinctions and extirpations were high (Fig. 2D)”.

2c) As a follow-up on 2a&b, expanding the predictions to +3C seems unfounded (e.g., Lines 228-230, Fig. 4), especially considering the limited time window studied and the poor fit of some of the models. Again, the argument feels circular to me, especially given the limited focus of the study (i.e., bivalves, brachiopods, gastropods, Early Jurassic, Europe) as well as the variable responses amongst taxa. This also assumes that species' niches and thermal tolerances cannot evolve, which should be directly stated in the text.

#+3 degrees regional warming is well within the empirical range here (see x-axis of Fig. 3), thus within the range of our model. We now added to the first Results sentence that the regional mean and SD over the individual warming phases were 4.5 ± 1.9 °C, which could have been another value for which to calculate our model expectation (which is really an empirical interpolation). We agree that we need to discuss the limitations set by the time window studied, location and taxonomic constraints and we have developed the Discussion of these and add a caution phrase at the end of the Results). However, a warming level needs choosing to set model expectations (predictions) to. Any warming level chosen is arbitrary (e.g. +1°C, +10°C, or the mean observed warming of +4.5°C), so we decided to make it a central but more modern-applicable value. We expect that future work will make more suitable predictions.

We agree that our approach does, like many, assume that species thermal tolerances tend to be conserved, and we thank the reviewer for noticing that we had failed to mention this (now supported with a reference, Saupe et al. 2015). Certainly, species thermal tolerances evolve, but we take the view of Jablonski et al. (2013) that these tend to be rare events.

Jablonski, D., Belanger, C. L., Berke, S. K., Huang, S., Krug, A. Z., Roy, K., ... & Valentine, J. W. (2013). Out of the tropics, but how? Fossils, bridge species, and thermal ranges in the dynamics of the marine latitudinal diversity gradient. *Proceedings of the National Academy of Sciences*, 110(26), 10487-10494.

3) In Ecological Niche Modeling (ENM), one of the important assumptions is that niches are species-specific, rather than at an assemblage or family level. I am not suggesting the authors toss out the entire study, but it seems like an ENM would have been a better choice for a robust analysis of this type (or at least would back up the assumption that thermal tolerance is the main driver of diversity patterns). In that same vein, it seems like the analyses of depth and substrate were an afterthought, although I could not access the supplemental data, so maybe these issues were explored sufficiently there.

#We completely agree that niches are species-specific (or even population specific, if genetic information is available), as they are assumed throughout this manuscript. We think there is misunderstanding over our definition of 'assemblage', which we have aimed to clarify in our editing e.g. now defined as '*... without any requirement of cohesion, as the species present in a given spatiotemporal unit*'. At no point do we assume an assemblage to be cohesive or have a single niche – an assemblage thermal bias is the simple statistical average of its component species, which each are assumed to have their individual temperature preference, but species can freely recombine into a new assemblage (which we can newly sample the thermal bias of). None of our analyses are family level.

We are not sure what the reviewer understands as an 'Ecological Niche Modeling (ENM)'. We are familiar with niche models as various approaches to statistically quantify a species environmental preference on one or more niche dimensions (e.g. a simple mean as we have done here is a simple but common approach) and then projecting the niche model using modelling like Maxent or boosted regression trees and we mention them in our manuscript (end of first Discussion paragraph)). These are often termed species distribution models or habitat suitability models. Such models are usually used for species-specific insights, with a strength in projecting to unobserved sites or habitats. This is not the goal in our manuscript, nor the other studies of thermal bias, STI or CTI that we introduce (Devictor et al. 2008; Stuart-Smith et al. 2015; Day et al. 2018). We estimate the thermal optimum of a species, use its proximity to a regional mean temperature and its observed regional occupancy response, and take the species-level relationship as the average over many species. Assemblage level relationships are similar but aggregate (mean) the species-level proximity to the regional mean temperature and the aggregated (%) observed regional occupancy responses, and take the mean relationship over multiple assemblages.

Again, apologies for omitting the supplementary, where several of the analyses of depth and substrate reside.

Other issues:

- Several acronyms were not defined (CIs, Cis), and explaining some jargon would make the paper more accessible (e.g., I am always surprised how few people are familiar with extirpation).

#We apologise that 'Cis' was an erroneous auto-correction of 'CIs', which we have now defined as 95% confidence intervals on first mentioning (and Fig. 3 caption, in case readers skim the figures). We agree with the reviewers and now define extirpation in the Introduction: "species extinction, both local (henceforth termed extirpation) and global (henceforth termed extinction)".

- Some sentences are awkward or I had a hard time understanding what the authors were saying (e.g., lines 64-66, 77-79, 126-129, 140-142, 222-227)

#Thanks for highlighting these specific sentences. We have amended all of the above and other points mentioned by all reviewers have helped throughout the manuscript, like avoiding the previous use of 'escalation' and overuse of statistical terminology.

- A few minor grammatical issues (e.g., lines 74, 475, 600)

#Amended.

- What are the numbers in Fig. 1A?

#Now added, "Values in the legend are total number of occurrences".

- I would have liked to see the climate data from each of the ammonite subzones. Maybe this was in supplemental.

#Yes, the global maps were indeed in the supplementary. We have now added another series of maps zoomed into the northwestern Tethys, which cover the range of temperature values of the main CO₂ scenario, from Spinatum at 400 ppm to Exaratum at 1000 ppm (new Fig. S4).

- How was a "change in habitat substrate" determined (line 142)? There is a lot of discussion about habitat, but no explanation of how it was determined; again, maybe this was in supplemental data.

#We admit this was hidden in that statement, "Substrate or bathymetric categories were combined using the keys in divDyn". We now add a SM Section 'Substrate and bathymetric categories', where this categorisation is made explicit, so an interested reader would not have to use R to find this information.

- Lines 147-151: Please show the results for the bivalves/brachiopods as well as all the data together. Without the data the claim is not substantiated.

#Apologies, these lines have now been clarified as referring to the interaction terms in Table 1, which demonstrate the difference. The effect of thermal bias on occupancy response for each group in separate models is in SM section 'Influence of number of species, facies, and clade on thermal bias – occupancy response relationship'.

- Line 201 needs a citation, again, niches should be assigned to species.

#References added.

- What do you mean by "the share" of a group (e.g., 237)? Please explain.

#Changed to 'proportion' in all cases.

- Line 253: "Escalating response gradient" This conclusion is not always founded based on your dataset.

#Qualified as 'during warming'.

- Lines 309-313: What data is this referring to? The results in Fig. 2 seem to disagree in places (especially concerning the immigrating/originating species).

#Added 'during warming and transitional phases'. These are the values in the second paragraph of the results, the justification for which has now been moved here (previously in the methods).

- Lines 415-417: This conclusion is not shown in the dataset, or at least I am sufficiently confused about when STI vs. CTI was used.

#To avoid confusion, this sentence, which refers to the species-level results (Table 1), is now moved to before mentioning the assemblage-level responses in the conclusion.

- Late, Middle, and Early Pliensbachian/Toarcian should not be capitalized as only the stages are official (i.e., it should be late Pliensbachian, or middle Toarcian, etc.)

#My mistake (Reddin) – these are now amended throughout main and supplementary.

- Line 542: define "sufficient occurrences".

#Added '(n > 25 per time step)'.

Reviewer #2 (Remarks on code availability):

The dataset was restricted to "users with access" so I could not view it.

#If the one pasted in the manuscript did not work, the reviewer-access link was also forwarded as a separate communication from the journal (e.g. reviewer 1 received it). Apologies if this was not the case.

Reviewer #3 (Remarks to the Author):

I was excited to review this paper, but I found it disappointing. The title does not accurately convey the contents of the work. The analysis considers a limited geographic region and taxa groups. The reasons for the selections of region, taxonomic groups and coarse temporal resolution used for the analyses are not clear to me.

#We are sorry the reviewer was frustrated with some aspects of our original communication of the work and we hope they agree the revised manuscript has improved in clarity, as aided by their comments, which we are grateful for. The title is changed now.

There are good reasons for the geographical and taxonomic focus, and for the temporal resolution, which is actually fine -- rather than coarse -- resolution for paleobiological analyses of such large geographical areas. We now hope to have made these reasons clear in the introduction (e.g. 'well-sampled' area, 'modern analogue' interval). We admit the previous terminology of 'escalation', which was in the title, was unintuitive, and has now been avoided throughout the manuscript.

The paper is written as though an important general empirical model is proposed, but it is based on data from three taxa groups in one small region of the world over one event. I did not find the results particularly interesting or useful, and they needed better embedding into the literature in terms of what we know about present and future climate change, or that occurring in the Toarcian.

#We are surprised the reviewer did not find our results particularly interesting or useful. The parameter of thermal bias was proposed by modern ecologists (Stuart-Smith et al. 2015) to quantify how the local occupancy of a taxon corresponds to its thermal niche, thus how occupancy changes as local mean temperatures change. Ecologists commonly assess that populations at the equatorward side of their

species range are vulnerable to warming. In other words, thermal bias makes a spatial expression of a species' temperature performance curve, which is often demonstrated in laboratory settings, and underlies the concept of species and community temperature indices (i.e. STI and CTI; Devictor et al. 2008; Stuart-Smith et al. 2015; Day et al. 2018; Flanagan et al. 2019; Bonachela et al. 2021). For the first time, we use paleoecological data as evidence that validates this metric, up to climate-change-associated global extinctions, which is not possible using modern time series.

Despite this being a really well-sampled region for the event, data are combined into coarse spatial and temporal units. The amount of change explained is modest at best (18%) and this is based on only a subset of the data used. Interpretations are also based on non-significant statistical results.

#We are unsure of the reviewer's frame of reference as we have very different viewpoints on several points here and below. We aim to use these points to clarify our messaging for a wider audience.

Aggregating occurrences into spatiotemporal units is standard practice in macroecology and paleobiology, alike, to help standardise sampling effort and scaling aspects (e.g. relationships between biology and environment usually change with scale of observation). High resolution units in paleobiology, such as sedimentary beds at an outcrop, often cannot be correlated (identified as the same sedimentary unit) over even short distances, so were not feasible for our study that covers 1000s km. Stratigraphy is the science of correlating outcrops, and the stratigraphic zone (sometimes subzone) is the finest resolution that can be correlated over 1000s km, thus was most practical for our study, being finer than the stratigraphic stage.

#(Paleo)ecological systems are complex and produce notoriously variable responses. An R-squared value of 18% for a single variable like temperature here is actually impressively high in paleo or modern ecology. The larger model, including variables other than thermal bias, has a value of 30%. We discuss two instances of coefficients with marginal significance, which are likely marginal because of a small sample size, in a wider evidence context. This reflects good statistical evidence-based practice.

Insufficient consideration is given to the importance of other environmental drivers, and the considerable influence of biotic factors on structuring populations, communities and ecosystems across broad spatial scales. The effects of species interactions seem to be almost entirely neglected, but they can be the primary driving factor (e.g. classic experimental marine ecology throughout the 70s, 80s, 90s, 00's).

#We disagree with the reviewer. We consider the influence of water depth and substrate changes on our assemblages and assess the influence on our results of species with a high fidelity to depth or substrate habitat. Classic experimental marine ecology focussed on much finer spatial resolutions than the 1000s km distance units studied here, for instance showing how biotic interactions tend to drive zonation patterns of organisms in the lower intertidal, while abiotic factors drive zonation in the

upper intertidal. In fact, there is a wealth of modern ecological evidence that species interactions in general are more important in driving species presence or absence at finer spatiotemporal resolutions, while abiotic factors, especially temperature, are more important over broader resolutions (Tittensor et al. 2010; Sunday et al. 2015). Nevertheless, we did look for indicators of biotic interactions, especially habitat forming taxa, but coral reefs are generally absent from this time interval and lithotid bivalve reefs thrived earlier in the Pliensbachian.

Almost no consideration seems to be given to uncertainties on species distribution data, taxonomic identification, its stratigraphic constraints and so on. The grouping by space and time is not well explained or reasoned. There is some very highly temporally constrained data available for the TOAE, but it is lumped into ammonite zones most of which are hundreds of thousands of years duration.

#We have aimed to make this clearer (especially in the Introduction last paragraph), but we took considerable effort over such considerations. Studied species distributions are of the better sampled two-timer species only, binned to large regions, and the relatively high R-squared values show uncertainties were well-constrained. We vetted our dataset to homogenise taxonomy and now add a table of change (before vs after) of the original identified and accepted species names from the PBDB (Table S11). We now also add a table of the stratigraphic binning to the supplementary (Table S12).

If this paper is supposed to be written for a general audience of Nature journal readers it is not effective, e.g. a non-geological audience would not know what an ammonite zone is or means, nor would they appreciate first and last appearance, or what the Signor-Lipps effect is. None of the concepts or approaches are referenced, e.g. those that are conceptual/artefactual (Signor-Lipps), analytical (Euclidean distances, cluster analyses).

#The reviewer is correct that these concepts should be accessible and we have added details where we thought necessary. However, we disagree that references are needed for some widely used concepts, especially Euclidean distances and cluster analyses (the package vegan is already referenced).

It lacks the polish or clarity I expect for a submission to this journal. The style of expression, choice of language, explanation of the approaches chosen and the assumptions made will not engage a broad audience.

I found the abstract and introduction vague and not always well-supported by the literature. Understanding of changes in species biogeography is poor, and over simplified.

#The above two points are largely similar and we thank the reviewer for highlighting sections that could be made clearer, which we have now done.

Introduction:

Overall, the literature cited is insufficient and is missing key contributions. Many key points are made citing no references, or using a single reference – this does not demonstrate a robust evidence base exists nor incorporate differing perspectives.

#NComms states “As a guide, references should not exceed 70”. Unfortunately, this limits how many references we could include. We had to be selective with those included to demonstrate the robust bases of evidence, especially given the interdisciplinary nature of our work, such as thermal bias references from the modern setting. References that supplied the most PBDB occurrence records that we used were also prioritised for main text inclusion based on their science, given that those were also the key data sources (i.e. the top references in Table S13). Further references are also provided in the Supplementary Materials text. We would be happy to incorporate differing perspectives if the reviewer can direct us to them.

L37 In general, BUT there are species that have expanded ranges, contracted ranges, shifted ranges, some that have deepened their distribution, and some that have not changed at all.

#The references supporting this statement analyse the variation in distribution changes among species, populations, etc. The trend of poleward range shifts, to which we refer, is clear e.g. accepted as ‘high confidence’ by the IPCC. It is not productive to mention every exception. We add ‘widespread’.

L38 Vague. the amount matters if you want to make comparisons modern and ancient

#Unclear whether this refers to “repercussions for human well-being and ecosystems” or “Warming is projected into the coming centuries”. Both are referenced and are not the focus of our MS, just the context.

L42 Meaning of “performance decline”?

#Now the sentence links to “physiological stress”, as example.

L47 more needed

#Suggestion is not clear, but sentence is amended.

L52 unclear

#Amended

L54 linked to species latitudinal shifts

#Changed to ‘associated with’

L57 Hypoxia and anoxia

#Here we mean anoxia. Specialist macroorganisms can still thrive in hypoxia.

L57 other toarcian refs needed to support this - but also from other time periods, and modern

#Again, we are at our reference limit. Please specify if the reviewer has key reference preferences.

L58 inferred/determined from? i think thats what you've done?

#Sentence changed.

L60 is optima the important component? or is maxima and minima that has deleterious effect, plenty of species occupy a realised niche not a fundamental niche

#Agreed (e.g. Stuart-Smith et al. use both thermal mid-points and tolerance limits, as do we e.g. Table S7 and 'Cold and hot thermal bias results'), although it is not the focus of this paper. Added 'and tolerance limits'.

L64 Thermal bias: the concept needs exploring more. I would expect species distributions to be defined by their thermal limits, i.e. the range of temperatures not the optimal. In either case where is this key assumption/interpretation explained?

#Added "and closer to tolerance limits". All of these values are simple indicators extracted from data, while in reality organisms' temperature tolerances depend on wider abiotic and biotic variables. The statistical relationships derived from these indicators are nevertheless useful.

L69 "the wider validity is rarely tested" so how can it be used with confidence to make meaningful interpretations for fossils?

#The idea is that fossils can provide such validity, especially for extinction-related observations.

L87 uncelar

L74 "Warm" is vague

#For clarity, we have now changed throughout to use 'positive' and 'negative' thermal bias, rather than 'warm' and 'cool'.

L76 "persisting species" needs defining

#Done in more detail in the methods i.e. Fig. 5.

L90 needs defining

#'Two-timer species' are now defined on first mentioning.

L91 Why only these taxonomic groups?

#Added more detail in new edits.

L102 what literature?

#Added Nordt et al. 2022 and McElwain et al. 2005 here. Explained in detail in the methods and supplementary section 'Climate conditions of our major time steps'.

L104 how do you derive a summer temperature from geological/palaeontological materials?

#Clarified as model-derived.

Was the analysis completed on species or genera? How were taxa classified as immigrating etc? Is the analysis on assemblages or species – what are the assemblages composed of?

#'Species' is mentioned here several times and genera/genus is not. Much of this detail is correctly in the methods section, where we have rewritten parts for clarity.

How were thermal niches determined, what are the assumptions?

#Thermal bias is defined in the third paragraph of the introduction.

Many assumptions are made in relation to the approach that are not explained, i.e. that biogeographic distribution reflects physiological ideals and limits I guess? There is very little explanation of the biological or ecological basis.

#This has been introduced throughout the introduction and details of our approach are in the methods.

The temporal resolution is very coarse. Some of these ammonite zones are thought to be of considerable duration– how does it match current day shifts?

#Discussed in the discussion.

Results

Regular use of “bins” is not helpful, why not just call them categories?

#These are nearly always qualified as “time bins” and mostly used in the methods, being used twice in the introduction and once in a figure caption in the results. ‘Category’ ignores the temporal aspect.

Makes a lot of reference to the supplement – should it really be a supplement?

#Because of page and figure limits, it has to be, but the key evidence is presented in the main text.

What is $p < 0.2$ taken to indicate, this is not a significance value used by any statistical analyses I have ever used. If it is to be used then it needs to be discussed and backed up by some discussion and evidence that it could be considered statistically meaningful.

#Agreed. Changed for $P < 0.1$, which is generally considered ‘marginal’ significance.

L201 “Faunal responses to climate change are often measured or projected at the level of assemblage.” Not in my experience, but you should cite your sources so that we can see what works you are referring to.

#References now added.

L221, 224 Why is interpretation made from analyses that are not statistically significant? Similarly non-significant trendlines are plotted on figures 2b, 2e, 2f, 3

#This comment refers to the comparison of two relationships on overlapping datasets, one with $P < 0.006$, one with $P < 0.058$. We believe, as do many statisticians, that the consideration of frequentist alpha levels as binary thresholds of “useful vs not useful” results is poor practice. Significant and marginal relationships should be considered as evidence in light of effect size, sample size, known biases, and further evidence. All slopes are plotted in Fig. 2 for readers to assess and interpretations take their P-values (significance levels) into account. Figure 3 is now simplified to a single regression line.

Fig. 1, surely temp change is more important than absolute temperature?

#CO₂ change is shown in Fig. 2A and values of temperature change are shown in a new supplementary figure (Fig. S4).

I did not find fig. 4 intuitive or easy to read

#Agreed. Fig. 4 is now overhauled.

Figure captions and labelling are unclear – see annotations on the text

The phases of warming, cooling and temporal stages etc are not sufficiently contextualised

#Suggestion unclear but several changes have been made to the ‘phases’ introduction (e.g. beginning of Results) and further queries are probably aided by access to the supplementary materials.

Modern geographic references seem to be given priority over palaeogeography in the descriptions which makes no sense given the movement of species would not have been determined by today’s geography. E.g. the Germanic basins, the British basins and so on.

#We are not sure what the reviewer wants here. ‘Germanic’ and ‘British’ basins are just geological terms in the literature corresponding to where the fossils are found today. Naturally, we use paleocoordinates for all analyses.

How have species absences been dealt with, how do you know a species is missing? There is much mention of varying sample size but how does that reflect the effort taken (or not) to establish that a species was absent? What about stratigraphic

or preservational biases? Some of the gastropods in these sections are very small.
#We use two-timer species, whose record of presences and absences may be less influenced by sampling fluctuations (now added to the end of the introduction), for greater confidence in patterns of regional species presence and absence. In the supplement are the additional analyses using three-timer, rather than merely two-timer, species, to assess its effect on the results.

Discussion

L249 “removal” meaning?

#Changed to ‘extirpation’.

L250 regional species loss = extirpation, but also insufficient use of literature, it is extensive

#It is. We added reference to some studies that are elsewhere cited in this manuscript. Again, please understand that we had to meet reference limits during the formatting.

L256 taxonomic membership?

#Changed to “the species’ taxonomic grouping as bivalve or brachiopod”

L260 Why?

#Line is now rewritten.

I found some of the interpretations inflated, e.g. line 252 “We present empirical evidence from the fossil record that immigration, persistence, extirpation, and likely the extinction of species form an escalating response gradient linked to species suitability to regional conditions, as estimated by their thermal bias.” Based on information on local sea surface temperature only, information that is inferred from proxies? The sources of data used to infer temperatures or thermal bias were not described or critiqued, authors are referred to the supplement for this but surely this is critical?

#This statement is exactly what our statistics show. It is not inflated. The seawater temperature data are from climate models that were supported or validated using local proxies. The climate models themselves have their own literature, some of which is cited here, where they are critically developed (Willeit et al. 2022, and references therein). Without putting a thermometer in a time machine, this is the best that we can do. We have tried to make it clearer that temperature data are from climate models (e.g. second-to-last sentence of Introduction).

If thermal bias or temperature tolerances are gleaned only from the fossil distributions themselves, how do we know that their distributions are explained only by water temperature?

#As discussed with reviewer 2, we are not claiming that regional occupancy responses (rather than distributions) are explained only by water temperature. But we do aim to show that modelled temperature, especially thermal bias, is an important predictor – not saying it is the only predictor – and the text has now been clarified as such. There are many modern ecology studies that evidence and focus on the importance of temperature on setting a species' distribution (several referenced here e.g. Tittensor et al. 2010). Sunday et al. (2012) shows how marine species range edges in particular tend to closely match their thermal tolerances. This serves as the background for the study of STI, CTI and thermal bias (Devictor et al. 2008; Stuart-Smith et al. 2015; Day et al. 2018; Bonachela et al. 2021) and our own study on fossils here.

L273 There is evidence for very large shifts in body size of up to 50% in the as a response to environmental change in the taxonomic groups, considered analysed in this study that are not cited. It is not simply a case of taxonomic groups of smaller size and life history, there is actually evidence from this event growth was stunted by some factor.

#We agree with the reviewer that there is evidence that growth was stunted, for instance in the case of the bivalve *Pseudomytiloides* in the Cleveland Basin (Morten & Twitchett 2009), and both growth and morphology of belemnite molluscs in Peniche (Nätscher et al. 2021), both of which we now reference. Although our manuscript focuses on regional species turnover rather than what happens within persisting species, we now mention within-population growth response.

L267 dispersal, so why haven't these factors been considered in this work? How is the connectivity of ocean basins, or species etc that might have facilitated changes in species biogeographic ranges (or not) considered? Sessile invertebrates have differing living habits and life history mechanisms, some are local dispersers only.

#Our work here is correlative but the reviewer is correct that mechanisms like dispersal will affect the correlations we observed. We discuss dispersal in this paragraph and throughout section 'Linking climate-driven range shifts to extinction risk', exploring the likely role of geographical and anoxic barriers to the north of our study extent. Much of the dispersal ability and other life history mechanisms of these organisms is unknown. Applying uniformitarianism and the direct evidence we have from fossilised larval shells of Jurassic bivalves, we can assume that the larvae of our focus groups had a pelagic phase (bivalves: planktotrophic and lecithotrophic larvae; rhynchonelliform brachiopods: lecithotrophic larvae; the few gastropods with potentially direct development are ignored in some of the quantitative analyses anyway e.g. Table 1). Owing to their pelagic phase they have good dispersal potential, larger in planktotrophic than in lecithotrophic larvae. However, given the long geological time spans involved, providing dispersal chances for a multitude of

succeeding generations, differences in larval strategy are unlikely to be a controlling factor. Furthermore, taxa in our study region thrived in epicontinental seas and did not have to cross a large open ocean. Rather, we suggest occupancy was a question of suitable living conditions (temperature included) and circulation patterns.

We have supplemented the text on this topic, noting how regional models (Bjerrum et al. 2001; Ruvalcaba Baroni et al. 2018) suggest clockwise NW Tethys surface currents had likely weakened by the time they reached the northern shelf, encouraging stratification and local anoxia.

Some basins were restricted and were not always well connected to other regions because of other factors of palaeogeography, habitat loss, food supply etc. Insufficient consideration is given to other influences on their recorded or likely predicted future distributions. Likewise, the success or failure of any changes in species distributions and possible extinctions are not determined only by temperature, biotic factors such as predation and competition are likely to be incredibly important, yet aren't considered at all. Again, any estimates of extinction vulnerability (L277) are only due to temperature.

#Though we agree many other factors will be interesting to study during this time and region, many have currently insufficient data, especially at the temporal and spatial scales here. We do not assume that temperature change is the only determinant of extinction, just that it is correlated here and all evidence suggests it is likely to have been important then as now in driving extinction risk (in a statistical rather than deterministic sense). As discussed above, biotic factors are more important at determining presence/absence at finer spatial and temporal scales than those observed here (e.g. local but not regional exclusion; see Flanagan et al. 2019 and references therein). Broad-scale biotic factors include biogenic reefs, which are too rare to study during this time and region. Furthermore, biotic dependencies are not well constrained in these Early Jurassic organisms. Competition is notoriously difficult to assess from the fossil record, particularly in our study system (level bottom communities of macroinvertebrates) with few exceptions (e.g. patterns of overgrowth in bryozoans). The inter-connectedness of regions has now been supplemented by the response to the previous comment.

L277-278 what do you consider to be “rapid warming”?

#Added ‘geologically’

L284 like the habitat loss that would have occurred during this event where anoxia and ocean acidification were key?

#Added, ‘including oxic habitat’.

L286 There is good evidence for range shifts in marine invertebrates (these are

vulnerable to warming) and some fish are not changing ranges, some are simply expanding and some aren't moving or are going to greater depth. E.g. see Poloczanska et al. 2013

#We agree, though the evidence for a general trend of poleward range shifts is better in bony fishes e.g. Poloczanska et al. 2013, which we cite often in this manuscript. Unclear what the reviewer wants adding.

L301 reference needed

#This is speculation based on our observations.

There is mention that L320 "Poor dispersal capabilities and/or dispersal barriers can lead to a species' failure to lessen its population thermal biases by shifting distributions, thereby shrinking its geographical range 48, and making it vulnerable to global extinction 45,49." So, why haven't these factors been considered in this work?

#As before, because they are unknown and, frankly, modelling fossil organism dispersal and its effect on extinction risk is outside the scope of our manuscript. We present and discuss a correlation between thermal bias and occupancy patterns. Dispersal limitation will contribute noise to that correlation, as we discuss here.

L322 "Mechanisms of and limitations to habitat tracking should be explored during other intervals with changes in climate, sea level, and geography e.g. 49." - Why not consider them in this study?

#We do consider them in this study, for our focal time interval. Unfortunately, much of these validation analyses are in the supplementary, which the reviewer did not have access to.

L327, 333, L380 unclear

#Amended

L338 misses key literature

#References are very limited. We added here another previously used elsewhere in the MS. If the reviewer has specific preferences please let us know.

L390 explain

#Added.

Citation style is quite unhelpful and uninformative, e.g. L380 "despite evidence often to the contrary." Followed by no explanation of the contradiction. "Temporal resolution is not a problem per se for the application of paleontological insights to modern issues" followed by no explanation, if you are going to comment on this (which you absolutely should) at least explain why. Nature readers are diverse, you

should be reaching a broad audience, especially if you want to reach any audience beyond geologists.

#These two statements are mentioned above and have now been amended.

L393 cite evidence for the “current biodiversity crisis”

#Added.

L411: How did you consider the rare component ?

#Not applicable. This paragraph is about how, while we focus on common species, rare species tend to be less understood in general in ecology, but particularly so in paleobiology .

L413 Of gastropods, bivalves and brachiopods

#Added.

Approach and methodology:

The methods are poorly presented with much methodological information missing. I have annotated a pdf with detailed comments in relation to this.

#Thank you. I have copied those comments below for responses. I also checked the pdf comments on other sections, which mostly overlap with the comments above, but there were some that did not, which have now also been addressed.

434. Most of the species range data for this event are at much higher resolution so why do this? needs explaining

#Refers to sentence, “We used the finest regionally-consistent temporal resolution for our occurrence data, the ammonite zone”. Because, as stated here, this is the finest temporal resolution that is regionally consistent. Slightly rewritten now.

445. short lived but impactful on fauna

#We don't doubt that, and we already discuss its lasting impacts, but its non-permanent impacts are beneath the scale of this study.

454. Ref doesn't provide evidence of palaeoenvironmental changes in nutrients, turbidity or salinity

#We cite Danise et al. 2013 because it discusses the impacts of these factors on the fauna of the Cleveland Basin.

458. cite sources for this

#We have now added Ruvalcaba Baroni et al. 2018 and Piazza et al. 2020.

460. is this pCO₂ scenarios?

#Now changed where $p\text{CO}_2$ makes more sense

463. on what material?

#Added "of well-preserved brachiopod shells"

465. depths

#Changed

466. form the same model?

#Yes, but this sentence is not talking yet about running any model, just forming the scenarios.

477. first mention - defined/assumed/interpreted for what and from what?

#Reworded

479. for the Toarcian?

#Added "for 180 and 185 Ma"

489. what are the $p\text{CO}_2$ concentrations for the scenarios based on?

#The various sources we used are dealt with in the first paragraph of section 'Seawater temperature maps', where this table (Table 2) is already cited.

497. so were they sp or genera?

#Reworded. Always species.

500. not clear what was done re latitudes nor how

#Reworded

505. documented in supp?

#Rather than leaving it in the code, we now add synonymous species names as a supplementary Table S11.

507. just say "entries"

#Reworded

513-4. meaning? and what is the quality of this information in the PalaeoDB?

#Reworded. The information is from the peer-reviewed literature.

517. what are the scenarios based on? what data, what assumptions?

#Reworded, given that we used the word 'scenario' for the $p\text{CO}_2$ conditions. This refers to the paleogeography settings to calculate paleocoordinates.

523. not clear why needed to cluster by space. first need to explain you constructed clusters and say how/why

#Justification now added to the Introduction last paragraph: "To help standardise sampling effort and scaling aspects, including approximating the spatial resolution of available climate model outputs..."

525. Were taxa groups distinguished?

#I don't understand why this is relevant.

526-7. needs explaining

#Thanks, now rewritten.

531-2. why?

#The rationale was there but now slightly rewritten.

532. its palaeoecological not ecological

#Changed

536. how many were removed?

#This would not be informative, given that the number of these smaller 'noise' clusters would depend on how dispersed the data were. Moreover, the aim of this step is simply as a check.

541. meaning? how did this effect what you did?

#We don't understand what the reviewer means. The paragraph goes into detail.

542. what is "sufficient"?

#Added $n > 25$ per time step

547. evidence?

#Added

553-4. why?

#Two-timer and three-timer species are commonly thought of as more reliable for calculating extinction rates e.g. Alroy 2008.

573. what is it? how was it determined?

#Added that this refers to Fig. 5.

577. how was the thermal niche determined and what are the assumptions related to this?

#Added signpost to next section, where this is addressed

589. this needs to come earlier

#OK, the thermal bias definition in the introduction was perhaps unclear as referring to our work too. We now mention it explicitly in the Introduction second-to-last (hypothesis forming) paragraph.

596. meaning?

#'Escalatory' is now removed in all cases.

661. several since 2013, use more recent IPCC reports

#We do use the most recent IPCC reports (e.g. references 2 and 5), but those do not model past the end of the century in a comprehensive way. To avoid this potential distraction, we now cite Lyons et al. 2022, which discuss this issue and include long-term model forecasts, rather than the 2013 IPCC report.

I didn't find the further detail in the supplement helpful in addressing my methodological queries

L433, 455, 458, 547 references?

#Added.

What is the source of the approach on thermal bias?

#Stuart-Smith et al. 2015, as cited on the first mention of 'thermal bias' in the introduction and in the methods.

Information is delivered in the wrong order, the basics of the data collation and analysis are not explained. How is information grouped prior to analyses and on what basis palaeontological/ecological/environmental is not clearly explained (if at all!). Key terms are not defined: what are you considering an "immigrant assemblage" to be, what is a persistent assemblage?

#These quoted terms are never used. An assemblage does not move cohesively, as we state. The information is delivered in the correct order (regional details, climate models, fossil occurrence data, how these are grouped, how the parameters are calculated, statistical analyses).

How are spatial assemblages defined – I assume this is on the basis of assumed physiological tolerances of those taxa in life. Is the analysis conducted on species? The ecological unit from which a distinct biogeographical range (and by inference physiological limits) might be inferred or is it based on genera? In the methods it says "occurrences initially had to be accepted at least at the genus level", but there is no follow-up statement that I can see.

#We hope the new edits help the reviewer as this information is all there. Analysis is based on species (accepted genera had their 'identified name' checked to see if they

too could be vetted here as species), which is now made explicit.

Writing style:

In many places the text is impenetrable to anyone other than the authors. Punctuation is poor. Sentences are long and technical (often 4-5 lines in length), Definitions (even the authors own that are introduced for just this work) are lacking. There is no explanation or discussion of the ecological meaning of some of them and the core assumptions on which the analysis is based.

#Long sentences have now been separated and we have taken onboard the other general critiques.

Some style of writing overly mechanistic, clumsy and non-intuitive, e.g. “Non-temperature habitat (and sampling) heterogeneity is likely to dominate at scales beneath our effective spatial resolution” what is “non-temperature” habitat?

#Understood. Sentence rewritten and another use of ‘Non-temperature habitat’ is now replaced.

The term ecology is frequently used in place of the correct term palaeoecology for much of the contents of the paper.

#Replaced when talking about paleoecological clusters, but often we do mean ecological (based on modern evidence). We have tried to make this clearer.

REVIEWERS' COMMENTS

Reviewer #1 (Remarks to the Author):

I did enjoy reading this MS by Reddin et al. again. I still believe this is a very valuable contribution representing an impressive amount of work. The MS remains relatively technical, but this is probably necessary to correctly convey the results of these complex analyses and the authors did great in revising the MS to make it much more accessible. I also appreciated the extensive supplementary information. I think the MS is now suitable for publication, which I definitely support.

My only concern regards the code hosted on Zenodo, which I suggest to update before publication (see dedicated section below).

Best regards,
– Alexandre Pohl

#We are sincerely grateful to Dr. Pohl for his constructive feedback, over which we enjoyed revising the manuscript, and it is very pleasing to read that he enjoyed the manuscript. We aimed to optimise the readability without losing any accuracy of description and the reviewer comments have collectively, greatly improved this.

Reviewer #1 (Remarks on code availability):

I ran the code, and it now runs well, but:

(1) figures are not saved on disk, which is not very convenient
(2) figures do not seem to perfectly match the revised MS. For instance, Fig. 4 is the original version, not the revised one (heatmap). The README also refers to the previous title.

I encourage the authors to update this Zenodo repository before publication of the MS.

#Thank you for pointing these issues out and apologies that I had overlooked them. Figures should now save to disk as pdfs and should match the manuscript versions, especially Fig. 4. I also updated the README.

Reviewer #4 (Remarks to the Author):

Dear authors, I am serving as a fourth reviewer assessing your manuscript. I see how your results are relevant in the paleontological context and, more importantly, in marine conservation. The STI/CTI approach has become extremely popular among conservation biologists, but evidence of its value in anticipating the future of biodiversity needs to be more extensive. Your study is a significant step forward. My

only criticism is aligned with the second comment of reviewer #2: STI values should be estimated for the species' entire geographic distribution (Burrows et al. 2019, Supplementary Table 2). I understand the difficulties explained by the authors in getting these values, but this should be stated as a potential shortcoming of the analyses. Other than that, I enjoyed your manuscript.

Burrows, M.T., Bates, A.E., Costello, M.J., Edwards, M., Edgar, G.J., Fox, C.J., Halpern, B.S., Hiddink, J.G., Pinsky, M.L., Batt, R.D. and García Molinos, J., 2019. Ocean community warming responses explained by thermal affinities and temperature gradients. *Nature Climate Change*, 9(12), pp.959-963.

#We are very grateful to reviewer 4 for stepping in to further the peer review of our manuscript, and are pleased they see the value in it, which is very close to our motivation. Based on their useful suggestion, we add the following sentence with the above reference to the second paragraph of our discussion: “Ideally, thermal preferences (e.g. STI values) should be estimated over a species' entire geographic distribution 30, though the patchy fossil record means this is rarely possible.”

Attachment

Paper summary:

Reddin et al. combine global climatic simulations and paleontological data to investigate the relationship between species thermal optima and their response to warming during the Early Jurassic (including the Toarcian OAE interval) on the European Tethys shelf. Paleontological data consist in a curated PBDB subset. Paleoclimate data correspond to CLIMBER-X results bilinearly interpolated to the resolution of HadCM3. The authors estimate the thermal preferences of brachiopods, bivalves and gastropods by extracting simulated ocean temperatures at their (paleo-) spatial-temporal locations, and then determine for each of the 1-Myr time slices considered, the difference between the thermal preference of the species and the temperature experienced by the species during the time slice. They show that the magnitude of this difference (or ‘thermal bias’) generally constitutes a good predictor of the response of the species to warming: origination, immigration, persistence, extirpation and extinction. They show that their conclusions stand when origination and extinction are excluded from the analysis and when the climatic scenario is varied in the model. Finally, the authors provide an extensive discussion of the limitations and implications of their results, notably in the context of the ongoing global climate change.

General comment:

I think that this manuscript overall constitutes a very interesting and robust work and would be a valuable addition to the field. It would be of interest to scientists from different communities, ranging from paleontologists to (paleo)ecologists and modern-day biologists. Therefore, I encourage its publication in *Nature Communications*. However, I think that the manuscript requires significant revisions to make it accessible. While the introduction, the discussion and the methods are well written, I think that the results are too difficult to follow and encourage the authors to significantly revise that part to make it accessible to a wide audience. I think that this is really a necessary step before considering the publication of this manuscript.

Main comments:

1. Title and wording.

- I only understood the notion of escalation relatively far into the manuscript, and thus think that it may help removing this concept from the title and abstract.
- In general, I think that the wording should be revised to be clearer. An example:
 - Lines 28–29: “the relationship was overridden by severe seawater deoxygenation”: I am not sure this is grammatically correct and think it is difficult to understand this sentence before reading the main text.

2. Missing parts. I could not find the (apparently extensive) supplementary information. I was also unable to access the code, since the data hosted on Zenodo is not publicly accessible. In particular, I would be interested in seeing maps showing the patterns described on lines 159–162: the northern part of the Tethys is the highest in latitude and thus the coldest – ok. I have difficulties understanding how these regions may undergo the smallest temperature increase during the first warming phase: I would encourage the authors to check, and explain the mechanism behind this unexpected (hence potentially interesting) response.

3. Suggestions to make the results clearer.

- I would suggest to avoid any code-like language and shortcuts and instead expand on these notions in the form of textual explanations; e.g.:
 - Line 73: “(regression: response~thermal_bias)”. Such phrasing does not make it easier (or pleasant) to the reader to understand what the authors mean.
 - Line 80: “(hypothesis A regression slope becomes steeper)”
 - What is R_{marginal} on line 139? I do realize my background in statistics is quite weak, but still think that the text should be assessable to a wide audience.
 - Table 1, in general, is very difficult to understand. Line contents are relatively obscure and column headers are not defined.
 - The same for Fig. 3: the label of the y axis is difficult to understand and not explained.
 - Please define the ‘mixed-effect model’ (and ensure a consistent spelling throughout, “effect” vs. “effects”).
- Possibly include a summary of key explanations that are only found in the Methods, such as the way that thermal bias is calculated and how the clusters/regions are used in the analyses. I did get a much clearer vision of what was done in this study after reading the Methods.
- In general, it would be good to better introduce the paragraphs and ensure better transitions. For instance, the authors may want to emphasize that they first work at the species level, and then look at assemblages. In particular, the caption of Fig. 3 suggests that this analysis already considers assemblages?
- I wonder whether Fig. 4 may not benefit from a major simplification, converted to a simple heatmap.

4. Suggestions of citations. These are really just suggestions since I don’t especially want to push citing my own work, but I feel that these 2 papers may fall exactly within the topic and thus provide the references here in case it would help:

- When discussing the potential, combined impact of temperature and ocean oxygenation, the authors might be interested in referring to this recent study (<https://www.science.org/doi/full/10.1126/sciadv.adg7679>) where we provide quantitative analyses of the impact of these environmental variables on extirpation and extinction rates simulated in response to global warming.
- Here (<https://agupubs.onlinelibrary.wiley.com/doi/full/10.1029/2018PA003394>) we compiled geological evidence for the north-south redox gradient in the Tethys and used general circulation models to try and provide explanations, notably focusing on the notion of salinity-driven stratification along the northern coast.

Minor and technical comments:

- Fig 1: “E., W. and N. Ibera are east, west and north of Iberia”. Here and in the methods, the authors refer to such clusters, but they are not shown on the map.
- Fig. 2: would it be possible to show the interval of T-OAE?
- Line 256: what does “taxonomic membership” mean here?

- The authors used bilinear remapping to interpolate their climate model output. Isn't this method inducing biases, notably creating temperature values that are not present in the original model? Wouldn't be the near-neighbor method more conservative?
- Line 486: I am not sure one can define a "best" estimate of climatic sensitivity, but this value is effectively within the range defined by recent models and not excessively lower than the multimodel mean of CMIP5 and CMIP6 models (<https://agupubs.onlinelibrary.wiley.com/doi/full/10.1029/2019GL085782>).